# Activities of visual cortical and hippocampal neurons co-fluctuate in freely moving rats during spatial behavior

Daniel Christopher Haggerty[1], Daoyun Ji[1,2]*

[1]Department of Molecular and Cellular Biology, Baylor College of Medicine, Houston, United States; [2]Department of Neuroscience, Baylor College of Medicine, Houston, United States

**Abstract** Visual cues exert a powerful control over hippocampal place cell activities that encode external spaces. The functional interaction of visual cortical neurons and hippocampal place cells during spatial navigation behavior has yet to be elucidated. Here we show that, like hippocampal place cells, many neurons in the primary visual cortex (V1) of freely moving rats selectively fire at specific locations as animals run repeatedly on a track. The V1 location-specific activity leads hippocampal place cell activity both spatially and temporally. The precise activities of individual V1 neurons fluctuate every time the animal travels through the track, in a correlated fashion with those of hippocampal place cells firing at overlapping locations. The results suggest the existence of visual cortical neurons that are functionally coupled with hippocampal place cells for spatial processing during natural behavior. These visual neurons may also participate in the formation and storage of hippocampal-dependent memories.

*For correspondence: dji@bcm.edu

**Competing interests:** The authors declare that no competing interests exist.

## Introduction

Spatial information is presumably encoded by hippocampal 'place cells', which fire predominantly when an animal is at one or a few specific places (place fields) of a given space (*O'Keefe and Dostrovsky, 1971*; *McNaughton et al., 1983*; *Wilson and McNaughton, 1993*). How these cells establish the location specificity has been a subject under intensive investigation. It is known that path integration of the information generated from self-motion is important for hippocampal place cell activity (*Hafting et al., 2005*; *McNaughton et al., 2006*). However, external sensory cues are equally important, if not more. In particular, visual cues play a pivotal role in place cell activities (*Muller, 1996*; *Knierim, 2002*; *Colgin et al., 2008*; *Ravassard et al., 2013*). For example, changing the proximal or distal visual cues alters firing rates and/or firing locations of place cells in a one- or two-dimensional space (*Lee et al., 2004a*, *2004b*; *Leutgeb et al., 2005*), and the rotation of a salient cue card on the wall of a symmetric open arena evokes an equivalent rotation of place field locations (*Muller and Kubie, 1987*).

How visual cue information contributes to place cell activities is unknown. Visual cues are presumably encoded and processed in the visual cortex, which is connected to the hippocampus via a multi-synaptic pathway involving the associational cortices including the entorhinal cortex (*Miller and Vogt, 1984*; *Vaudano et al., 1991*; *Lavenex and Amaral, 2000*; *Furtak et al., 2007*). Computational models have shown that hippocampal cells can integrate input from cortical cells, in particular those encoding visual cues, to form a conjunctive response that is location-specific (*de Araujo et al., 2001*; *Schonfeld and Wiskott, 2015*). Alternatively, it is proposed that local visual cues influence place field location through entorhinal 'border' cells, whereas distal cues influence place field orientation via entorhinal 'head direction' cells (*Knierim and Hamilton, 2011*). Obviously, these models and others similar require the functional interaction between those visual cortical neurons encoding visual cues

**eLife digest** The brain is able to create and maintain a map of our surroundings as we go about our daily activities. It is thought that some of this spatial information is first processed in a region of the brain called the visual cortex, the information is then relayed to a region called the hippocampus, where the map is reliably stored. However, researchers do not fully understand how the brain transfers spatial information between these two regions.

To explore what happens to the information, Haggerty and Ji recorded electrical signals from the brains of rats that were being trained to run back and forth along a C-shaped track. As they ran on the track, the rats gradually developed spatial maps of there surroundings. Electrodes were used to record brain signals from both the visual cortex and the hippocampus as this development took place.

Similar to previous studies, analysis of the recordings showed that a specific population of neurons in the hippocampus, called CA1 neurons, produced an electrical signal whenever the rat ran past specific locations on the track, such as corners. Building on this Haggerty and Ji showed a population of neurons in the visual cortex called V1 neurons also produced location-specific electrical signals at the same time. Moreover, the electrical signals of these two populations of neurons fluctuate together in a coordinated fashion.

The results of Haggerty and Ji support the idea that there exists a specific population of visual cortical neurons that communicate with hippocampal neurons in the development of a particular spatial map.

and corresponding hippocampal place cells. However, experimental data for such an interaction are very limited (*Ji and Wilson, 2007*), and the nature of this interaction during natural spatial behavior remains unexplored.

To probe the visual cortical—hippocampal interaction during spatial processing, we simultaneously recorded cells in the primary visual cortex (V1) and in the CA1 area of the hippocampus, while freely moving rats traversed back and forth on a track. We aimed to test a hypothesis that there exists a specific population of visual cortical neurons that responds to visual features of the track with specific activity patterns and functionally interacts with those hippocampal place cells representing the same track. We found that, like CA1 place cells, a large number of V1 neurons also displayed firing activities that were specific to particular locations on the track, especially around the visual landmarks. The location-specific activities of V1 neurons, on average, led those of CA1 place cells in both the spatial and temporal domains. For those V1 and CA1 cells that displayed spatially overlapping activity, their precise firing characteristics co-fluctuated every time the animal traveled through the track. These findings suggest the existence of specific V1 activities that participate in the encoding of external spaces. Given the critical role of the hippocampus in spatial memories (*Scoville and Milner, 1957*; *Squire, 1992*; *Eichenbaum et al., 1999*; *Burgess and O'Keefe, 2003*), these visual cells may also participate in the formation and storage of the spatial memories encoded by hippocampal place cells.

## Results

We used tetrodes to simultaneously record firing activities of V1 cells and CA1 place cells in 15 freely moving rats, while they were performing a track running task for food rewards. During the task, the food-deprived rats were free to run along a novel C-shaped track (*Figure 1A*). Each end of the track (food well) was baited with a milk reward which, if consumed, was not refilled until the rat had run to the opposite end of the track to consume the other reward. Over time rats became habituated to the task and repeatedly travelled back and forth along the two overlapping trajectories in order to maximize the number of rewards. The recording started on the very first day (Day 1) and continued for 2–14 days, with 20–60 min each day. Eight animals reached at least Day 7. We quantified the task performance of all the rats from Day 1 to Day 6, and a Day 7+, which included those days ≥ Day 7. Behavioral performance was measured by the running speed and the number of times (laps) per minute an animal traveled through each of the 2 trajectories. Both parameters increased rapidly during the early few days of running and reached a stable level on Day 3 (*Figure 1B,C*), indicating that much of the behavioral changes occurred during the first few days.

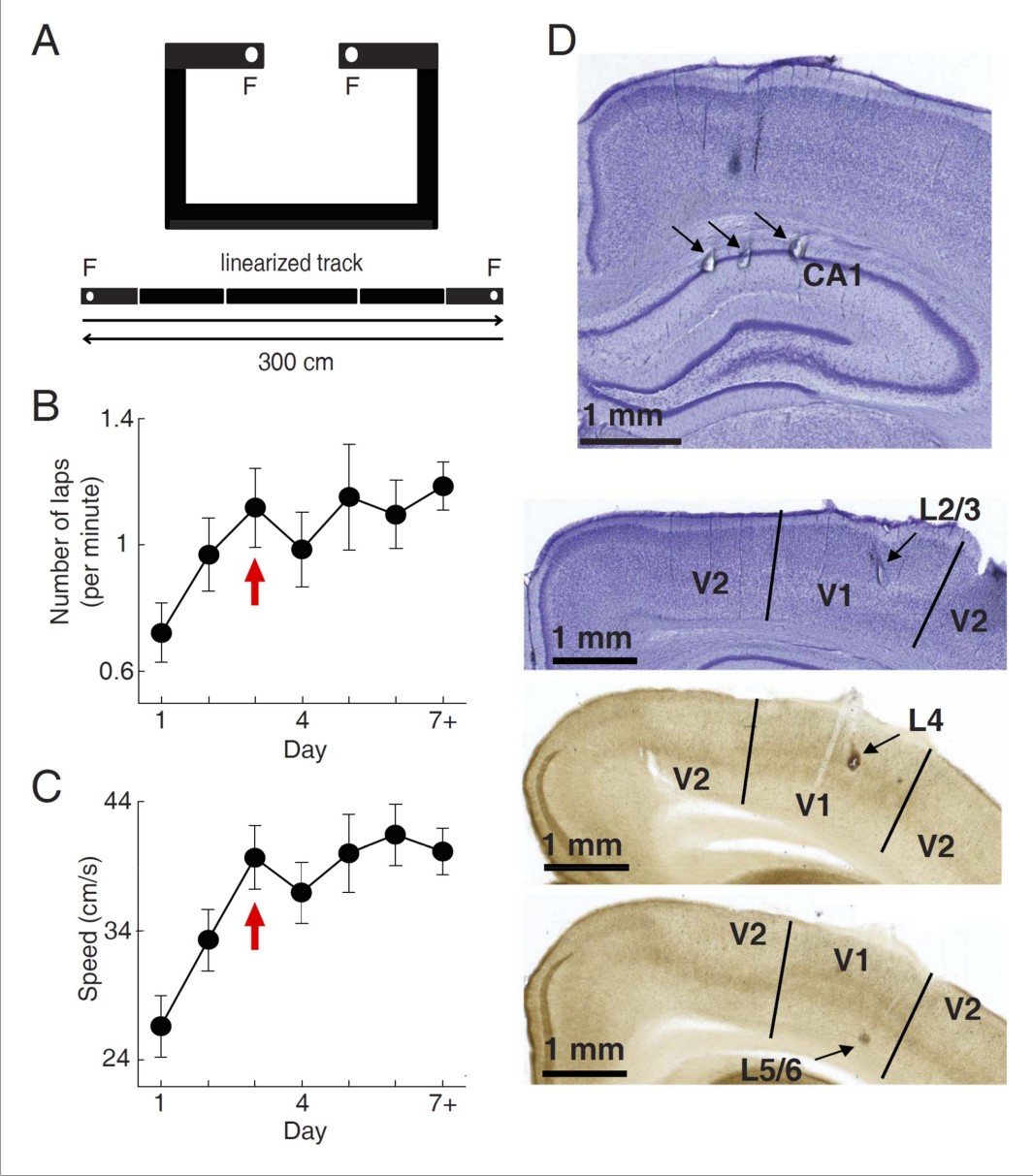

**Figure 1**. Behavioral task and recording sites. (**A**) A C-shaped track where rats ran back and forth for food rewards. *Bottom*: two linearized trajectories, each 300 cm long. *Vertical white lines*: corners of the track. *F*: food wells. (**B**, **C**) The mean number of laps per minute (**B**) and mean running speed (**C**) on each day of track running, averaged over all trajectories and all animals. The number for Day 7 + included those days ≥ Day 7. *Arrows*: the day when the mean number of lap and speed were stabilized. Number of trajectories: $N = 30, 26, 22, 22, 18, 16, 26$ for Day 1 to Day 7+, respectively. (**D**) Nissl- (top 2 sections) and AChE-stained (bottom 2 sections) coronal brain sections to show recording sites (*arrows*) in the hippocampal CA1, and those in different layers of V1 (L2/3, L4 and L5/6). V2: the secondary visual cortex.

A total of 852 V1 and 3627 CA1 cells were recorded across all recording days and from all the 15 animals ($10 \pm 1$ V1 cells, $44 \pm 2$ CA1 cells per day per rat, $N = 83$ day × rats; see 'Materials and methods' for details). V1 cells were sampled in all layers (*Figure 1D*). In this study, we analyzed 776 V1 cells and 2033 CA1 cells that were active on at least one trajectory (mean firing rate >0.5 Hz). Putative CA1 interneurons (mean rate > 5 Hz) were excluded. For each cell active on a given trajectory, we computed the overall firing rate of the cell during the running of the trajectory. The median firing rate

of V1 cells was 4.3 [2.0 9.9] Hz (median and [25% 75%] range values, $N$ = 1501 cell × trajectories), whereas the median rate of CA1 cells was 1.6 [0.97 2.5] Hz ($N$ = 2909).

## Location-specific firing activities of V1 cells

To test our hypothesis, we asked whether V1 cells responded to particular locations on a given trajectory of the track in a manner similar to CA1 place cells, with the assumption that the V1 response might not be a spatial response per se, but possibly resulted from the visual cues. As expected, CA1 cells exhibited typical place-field firing characteristics on the track (*Figure 2A*). Many V1 cells also dramatically increased their firing rates at specific locations of a trajectory (*Figure 2B–D*), as we have shown previously (*Ji and Wilson, 2007*). The location-specific increase in firing activity of V1 cells was apparently stable during each lap of the trajectory. The firing rate curves, defined as the lap-averaged firing rate at every position of a trajectory, of these V1 cells displayed a few well-defined peaks, similar in character to the place fields of CA1 cells and to the multi-peak firing of medial entorhinal grid cells on linear tracks (*Hafting et al., 2008*). Henceforth, we refer to the locations corresponding to the rate curve peaks of a cell as its 'firing fields'.

We quantified the location-specificity of each individual V1 and CA1 cell active on a trajectory. To compare with the location-specificity arising randomly from chance, we randomly shuffled the cell's spiking activity by circularly shifting the spikes within each lap of the trajectory with a random time interval (*Henriksen et al., 2010*; *Igarashi et al., 2014*). First, we computed spatial information content (SIc), a measure of how much information (in bits per spike) a cell's spiking activity contained about the animal's location (*Skaggs et al., 1993*). Although the median SIc value of V1 cells (0.17 [0.085 0.34] bits/spike, $N$ = 1501 cell × trajectories) was relatively small, compared with that of CA1 place cells (1.6 [1.1 2.2] bits/spike, $N$ = 2909; $p < 0.0001$, *ranksum* test), it was significantly greater than that of the shuffled V1 cells (0.061 [0.025 0.13] bits/spike; $p < 0.0001$; *Figure 2E*). Second, we computed spatial information rate (SIr), which measures spatial information in bits per second. Similarly to SIc, the median SIr of V1 cells (0.70 [0.42 1.2] bits/s) was smaller than that of CA1 cells (2.4 [1.2 4.2] bits/s; $p < 0.0001$), but significantly greater than that of the shuffled V1 cells (0.22 [0.14 0.37] bits/spike; $p < 0.0001$; *Figure 2F*). Third, using a method modified from previous studies (*Henriksen et al., 2010*; *Igarashi et al., 2014*), we derived a normalized spatial modulation index (SMI). The reason for this additional measure was that SIc and SIr are affected by firing rate (*Figure 2—figure supplement 1*). Since V1 and CA1 cells had different firing rates, the SIc and SIr values between V1 and CA1 cells were not directly comparable. SMI was defined as the SIc (or equivalently SIr) of a cell relative to its chance-level distribution produced by the random shuffling (*Figure 2—figure supplement 1*). SMI does not directly quantify the location-specificity of a cell's firing activity, but provides a measure of the degree of location modulation relative to random spike trains with identical firing rate and temporal spiking patterns. SMI is therefore insensitive to firing rate. The chance-level of SMI for any given cell is zero. The median SMIs of both V1 (8.1 [3.4 15.4]) and CA1 cells (12.0 [3.3 23.2]) were much higher than zero ($p < 0.0001$, *ranksum* test; *Figure 2G*). Finally, we defined a cell with SMI >2.325 (99th percentile of the chance-level) as a 'location-responsive' V1 cell. We found that 81% of trajectory-active V1 cells were location-responsive on a trajectory and for comparison, 90% of the trajectory-active CA1 cells were location-responsive.

Next, we attempted to understand why location-responsive V1 cells had lower location-specificity than CA1 place cells. One obvious reason is that V1 cells, unlike CA1 place cells, often contained much baseline firing outside of a concentrated firing field (*Figure 2B–D*). A second observation is that V1 cells often fired at multiple firing fields on a trajectory (*Figure 2B–D*), whereas many CA1 cells only exhibited one or two fields on our track (*Figure 2A*). We quantified the number and average spatial length of firing fields for location-responsive V1 cells. On average, the location-responsive V1 cells displayed nearly twice the number of firing fields as the CA1 place cells (mean ± SE for V1: 2.9 ± 0.04 per trajectory, $N$ = 1216 cell × trajectories; CA1: 1.5 ± 0.02 per trajectory, $N$ = 2618; $p < 0.0001$, *t*-test; *Figure 2H*). On the other hand, the average field length of V1 cells was significantly smaller than that of CA1 cells (V1: 37.5 ± 0.3 cm, $N$ = 3521 fields; CA1: 47.2 ± 0.4 cm, $N$ = 3918; $p < 0.0001$; *Figure 2I*). The result shows that the relative low location-specificity in V1 cells was partially due to more firing fields, but not due to larger field sizes.

We also quantified the lap-by-lap stability of a V1 cell's location-specific activity as an animal traveled the same trajectory repeatedly. For each cell active on a trajectory, we computed its firing

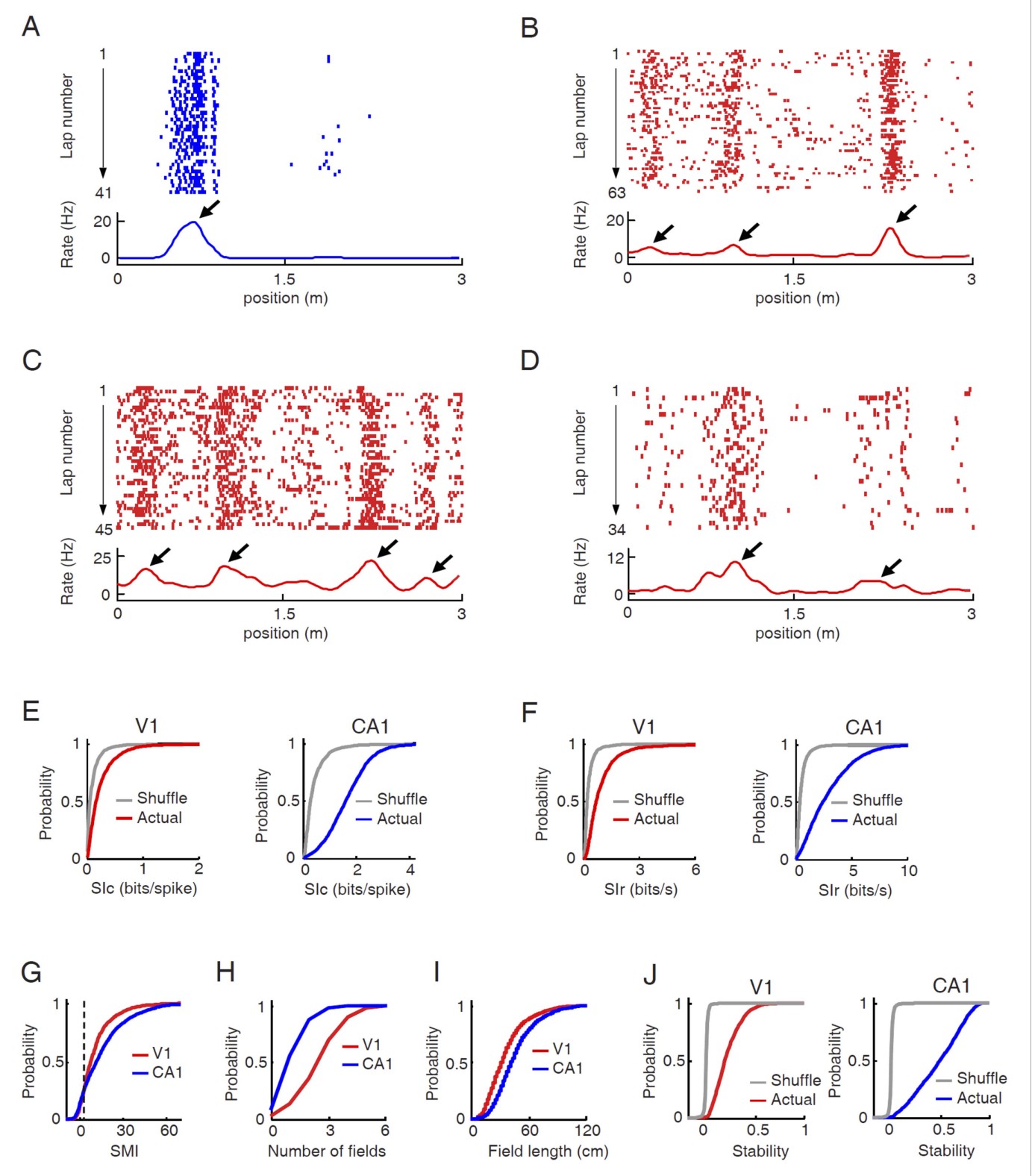

**Figure 2**. V1 cells fired predominantly at specific locations during track running. (**A–D**) Firing activities of a CA1 place cell (**A**) and 3 V1 cells (B , C, D in layer L2/3, L4, L5/6 respectively). For each panel, the *top* displays the spike raster of a cell within every lap of running on a trajectory, which is linearized and plotted as the x-axis. Each tick represents a spike. The *bottom* is the firing rate curve averaged across all the laps. *Arrows*: firing fields. (**E**) Cumulative distributions of spatial information content (*SIc*) values of V1 and CA1 cells with actual and shuffled spiking activities. (**F**) Same as E, but for spatial

*Figure 2. continued on next page*

*Figure 2. Continued*

information rate (*SIr*). (**G–I**) Cumulative distribution of spatial modulation index (*SMI*, **G**), number of fields per trajectory (**H**), and field length (**I**) for V1 and CA1 cells. (**J**) Same as E, but for spatial stability.

The following figure supplement is available for figure 2:

**Figure supplement 1**. Illustration of computing spatial modulation indices (SMIs) for two example cells with different firing rates.

rate curve separately for each individual lap on the trajectory and then computed the correlation between the rate curves of any two laps (*Cheng and Ji, 2013*). Spatial stability was the average correlation among all combinations of laps. To understand how the stability compared with the chance level, we also computed the spatial stability of each cell after the random shifting of its rate curves lap by lap. Similarly as in spatial information measures, the median spatial stability of V1 cells (0.19 [0.06 0.31], $N = 1501$ cell × trajectories) was lower than that of CA1 cells (0.51 [0.28 0.70], $N = 2909$; $p < 0.0001$, *ranksum* test), but significantly greater than the chance level (0.0014 [-0.0074 0.0060]; $p < 0.0001$; *Figure 2J*), indicating that V1 cells responded reliably to same locations during each lap of track running.

## Spatial correlation between V1 and CA1 firing activities

We next asked how V1 and CA1 firing activities interacted. First, we analyzed how the V1 and CA1 firing fields were distributed along the two trajectories on the track with opposite running directions. We found that V1 and CA1 fields were not equally distributed along the trajectories, but tended to increase in number close to the food wells, as shown previously for CA1 cells (*Dupret et al., 2010*). V1 and CA1 field numbers also increased before and after the animals traveled through the corners of the track, referred to as landmarks, and decreased as the animals traveled through the long segment between the two middle corners (*Figure 3A,B*). The numbers of V1 and CA1 fields along the trajectories fluctuated together in a highly correlated fashion for both trajectories (*Figure 3C*). We quantified this observation by computing the cross-correlation between the field distribution curves of V1 and CA1 cells on each trajectory. The cross-correlograms for both trajectories had peaks not exactly at position lag 0, but at a positive position (2 cm, $p < 0.0001$, *Pearson's r*) on one running direction and a negative position (−4 cm, $p < 0.0001$) on the opposite direction (*Figure 3D,E*), meaning that the number of V1 fields fluctuated ahead of the number of CA1 fields consistently on both running directions. Given the mean speed of 35 cm/s of our rats, the 2–4 cm leading distance is equivalent to 60–120 ms of leading time. While the higher numbers of firing fields around the food wells and corners might simply reflect more sensory information available around the landmarks, the significant correlation between the field distribution curves of V1 and CA1 cells with a spatial delay suggests an interaction between these groups of cells, possibly reflecting the propagation of the landmark-related sensory information from V1 to CA1.

The bias of V1 and CA1 fields around the landmarks on both directions prompted us to examine the directionality of V1 and CA1 firing fields. We observed that some V1 cells fired in front of or behind the landmarks on two opposite running directions, which we termed 'prospective' or 'retrospective' firing as in a previous study (*Battaglia et al., 2004*). An example of V1 cell with prospective firing is shown in *Figure 4A*, which was quantified by a significant peak at a positive position lag (36 cm, $p < 0.0001$, *Pearson's r*) in the cross-correlogram between the cell's firing rate curves on the two trajectories. An example of CA1 cell with retrospective firing is shown in *Figure 4B*, with a significant peak at a negative position lag (−18 cm, $p < 0.0001$) in the cross-correlogram between its firing rate curves on the two trajectories. We found that among all trajectory-active cells, 9.7% of V1 cells ($N = 75$) and 10.4% of CA1 cells ($N = 212$) were bidirectional, defined as those with a significant peak within [−50, 50] cm in their rate curve cross-correlograms (see 'Materials and methods'). The cross-correlogram peak positions of these bidirectional V1 cells display a bimodal distribution with two groups around 26 cm and −26 cm (*Figure 4C*), which corresponded to prospective and retrospective firing, respectively. Similarly, the peak positions of CA1 cells also show a bimodal distribution with two groups around 18 cm and −28 cm (*Figure 4D*). This result suggests that a group of V1 cells responded to the same landmarks on both directions by either 'looking' ahead or back to the landmarks.

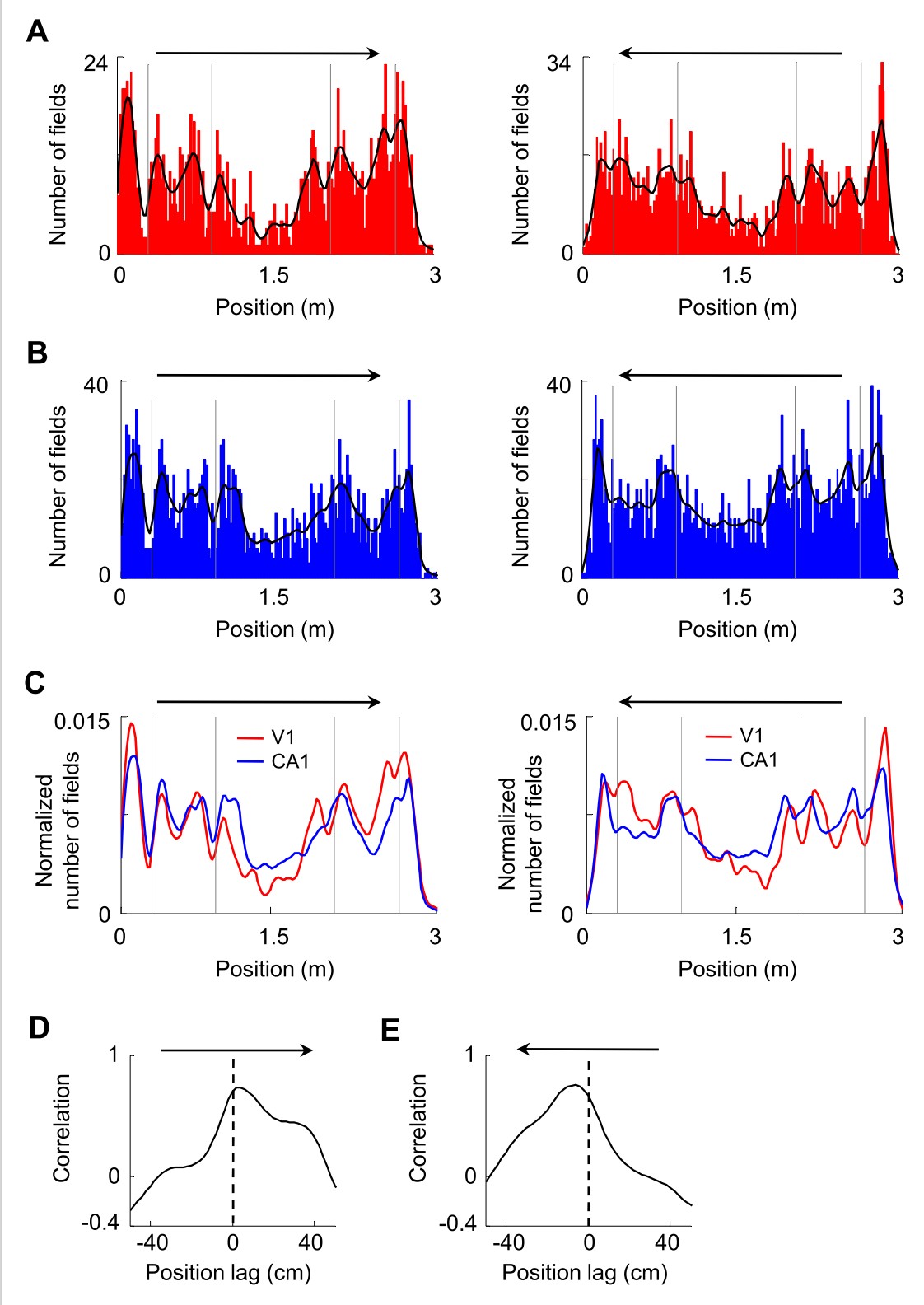

**Figure 3**. Firing fields of V1 and CA1 cells appeared to accumulate around the landmarks of the track. (**A**, **B**) Histograms of the number of firing fields for V1 (**A**) and CA1 (**B**) cells on the two trajectories of the track with opposite running directions. *Vertical gray lines*: landmark positions (corners) of the track. *Arrows*: running directions. *Black lines*: smoothed curves of the histograms. Note that the number of fields tended to peak before and after the landmarks for both running directions and for both V1 and CA1 cells. (**C**) The smoothed curves for V1 and CA1 cells in A and B are normalized (by total number of
*Figure 3. continued on next page*

*Figure 3. Continued*

firing fields) and re-plotted together for each of the running direction (*arrow*). Note that the V1 curve (*red*) tended to rise and fall slightly earlier than the CA1 (*blue*) curve on both running directions. (**D**, **E**) Cross-correlogram between the V1 and CA1 smoothed field curves in C on each of the running directions (*arrow*).

## Temporal correlation between V1 and CA1 firing activities

We then analyzed the pair-wise cross-correlation between V1 location-responsive cells and CA1 place cells in the time domain. We used a normalized spike count cross-correlation (see 'Materials and methods'), which is insensitive to the cells' firing rates, to quantify how the spiking activities of two cells were temporally correlated. *Figure 5A–C* shows an example pair of V1 and CA1 cells with both having well-defined firing fields on the same trajectory, which led to a prominent peak in its normalized cross-correlogram. The peak time quantified the temporal relationship of the two cells. For individual pairs of cells, this peak, especially that with a long peak time, might just passively reflect the fact that they had firing fields on the same trajectory, not necessarily reflecting any functional interaction. However, if a large number of such pairs are collected, the distribution of their peak times may inform the temporal relationship between the V1 and CA1 activities at the population level. Among 22,969 pairs of V1 location-responsive cells and CA1 place cells, we obtained 997 highly significantly correlated pairs (see 'Materials and methods' for definition) with peak correlation times within $[-0.2, 0.2]$ s. The distribution of these peak times was significantly biased toward positive values (58% positive, 46% negative, $p = 0.00017$, *binomial test*; *Figure 5D*). The result indicates that, on average, V1 cells led CA1 cells in the time domain, suggesting the propagation of visual information from V1 to CA1. We also analyzed whether this direction of interaction was true to those V1-CA1 cell pairs with both showing bi-directional firing. Out of the 12 pairs with both the V1 and CA1 displaying prospective firing and 9 pairs with one displaying prospective firing while the other displaying retrospective firing (only 1 pair with both displaying retrospective firing), 9 (75%) and 6 (68%), respectively, showed positive peak times (*Figure 5E*), suggesting that the V1-CA1 interaction of these pairs followed the general trend at the population level.

## Lap-by-lap co-fluctuation of V1 and CA1 firing activities

As many pairs of V1 and CA1 cells displayed overlapping firing fields on a trajectory (*Figure 6A*; see more examples in *Figure 6—figure supplement 1*), we reasoned that these pairs of V1 and CA1 cells could be specifically interacting for integrating the visual information present at a location to the hippocampal place cell activity encoding the same location. Therefore, we next focused on the possible interaction between V1 and CA1 cells with overlapping firing fields. We observed that the firing rates and firing locations of such V1 and CA1 cells fluctuated from lap to lap within their respective firing fields, and interestingly, they often fluctuated together in a correlated fashion (*Figure 6A,B*, *Figure 6—figure supplement 1*). We quantified this observation by an analysis similar to the 'noise' correlation in previous studies on V1 cell correlation in primates (*Zohary et al., 1994*; *Ecker et al., 2010*; *Hansen et al., 2012*). For a pair of cells with overlapping firing fields, we identified each cell's spikes within its firing field for each individual lap, and computed two quantities of these spikes: the (within-field) firing rate and the center of mass (COM) of their firing locations. We then obtained the lap-by-lap fluctuations in within-field firing rate (Δrate - the difference between a lap's firing rate and the averaged firing rate across all laps) and in COM (ΔCOM) for each cell. Finally, we computed the Pearson correlation in Δrate and in ΔCOM between two cells, which measures how closely the two cells varied their precise firing rate or firing location each time the animal traveled through the overlapped firing fields. Indeed, the pair of cells shown in *Figure 6A* were significantly correlated in both Δrate and ΔCOM (*Figure 6C*; see more examples in *Figure 6—figure supplement 1*).

In our data, we found 786 pairs with overlapping firing fields (see 'Materials and methods'). We referred to these pairs as 'overlapping pairs'. We found that 48% and 23% of the overlapping pairs were significantly correlated in Δrate and ΔCOM ($p < 0.05$, *Pearson's r*), respectively. As a group, the average correlation of all the overlapping pairs were significantly greater than 0, the chance-level correlation, for both Δrate ($0.29 \pm 0.01$, $p < 0.0001$, *t-test*; *Figure 6D*) and ΔCOM ($0.16 \pm 0.01$,

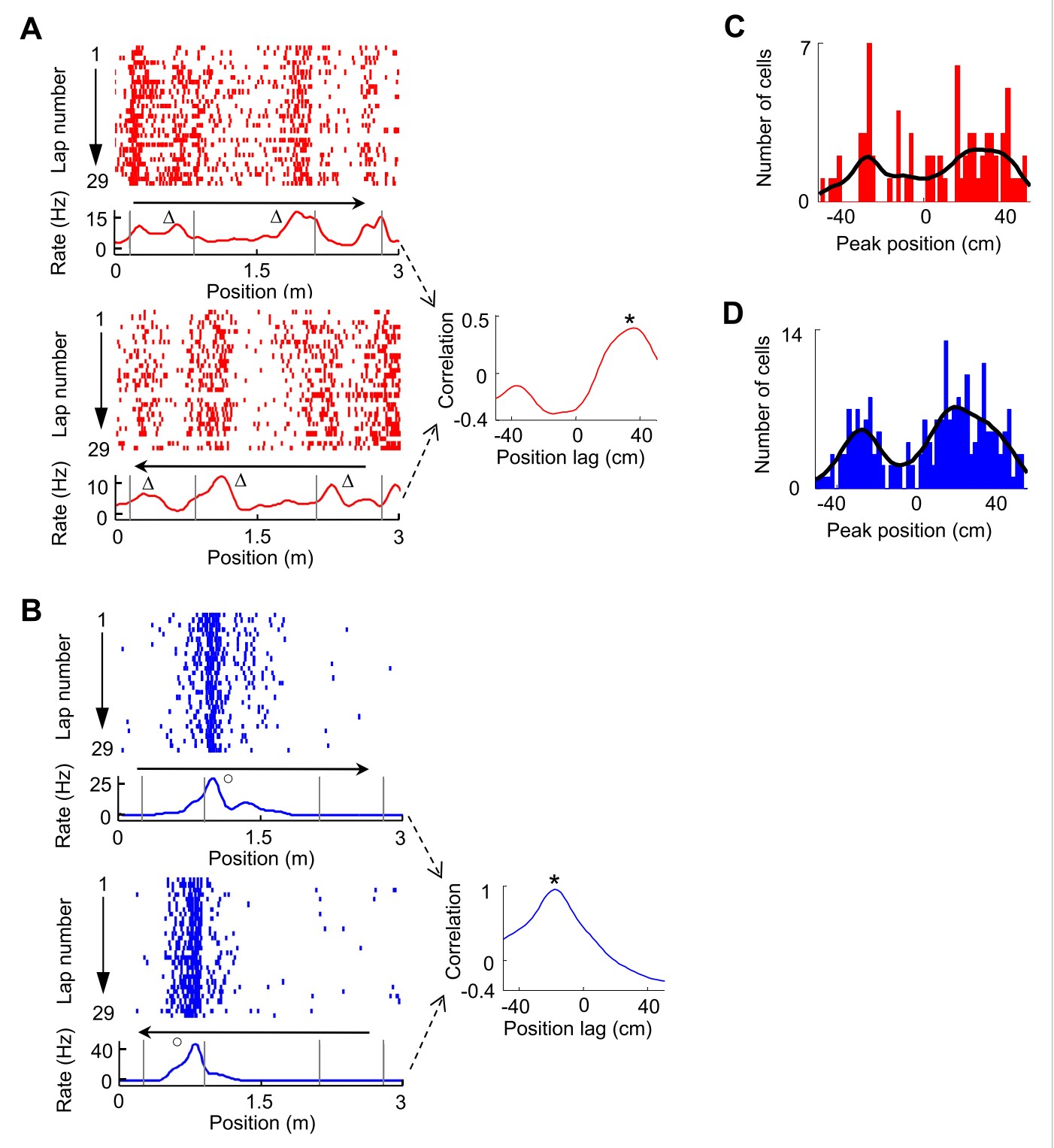

**Figure 4**. Bi-directional firing of V1 and CA1 cells on the C-shaped track. (**A**) Firing activity of a V1 cell (lap-by-lap spike raster and average firing rate curve; see **Figure 2** legend for details) on two trajectories with opposite running directions (*left*) and the cross-correlogram of the cell's two firing rate curves (*right*). *Vertical Gray lines*: land mark positions (corners) of the track. *Arrows*: running directions. Note that the peaks appeared before the animal passed the same landmarks (Δ, prospective firing) on both running directions, resulted in a primary peak (*) at a positive position lag in the cross-correlogram. (**B**) Same as A, but for an example of CA1 cell showing consistent firing after a landmark (o, retrospective firing) on both directions. (**C**, **D**) Histograms of the cross-correlogram peak positions of all V1 (**C**) and CA1 (**D**) cells with significant bi-directional firing. *Black lines*: smoothed curves of the histograms. Note that both the V1 and CA1 distributions appeared to be bi-modal, suggesting prospective or retrospective firing for V1 and CA1 bi-directional cells.

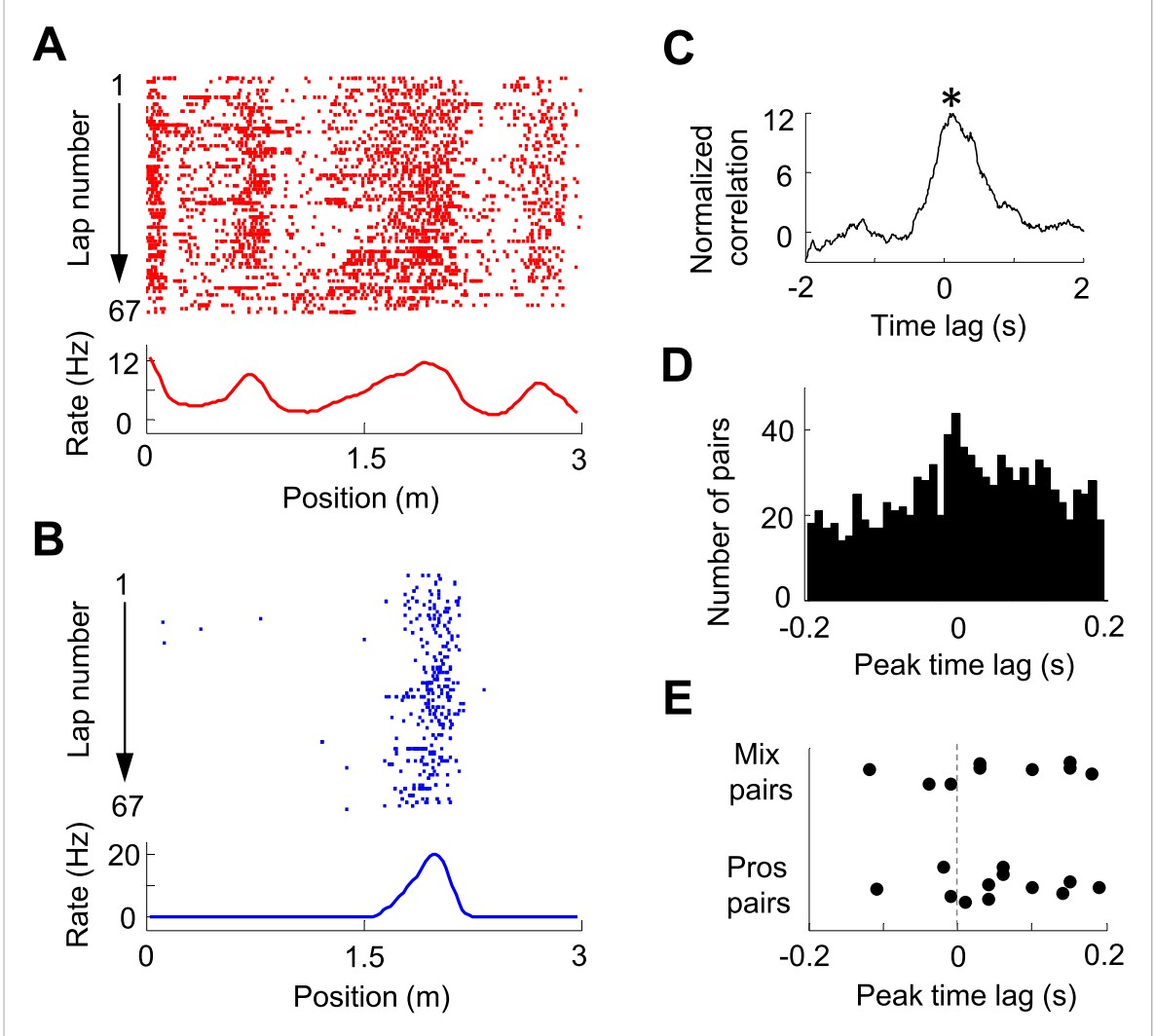

Figure 5. Pair-wise cross-correlation between V1 and CA1 cells. (A, B) Firing activity (lap-by-lap spike raster and average firing rate curve; see *Figure 2* legend for details) of a pair of V1 (A) and CA1 (B) cells on a trajectory of the C-shaped track. (C) Normalized cross-correlogram of the two cells in A and B. *: peak time of the cross-correlogram. (D) Histogram of the peak times for all highly significantly correlated pairs of V1 and CA1 cells (see 'Materials and methods'). Note the bias of peak times toward positive time lags. (E) Peak times of those highly correlated V1-CA1 pairs with both displaying prospetive firing (Pros pairs) and of those pairs with one displaying prospective while the other displaying retrospective firing (Mix pairs). Each dot is a pair.

p < 0.0001; *Figure 6D*). This result indicates that there was a precise lap-by-lap co-fluctuation in firing rate and COM between many pairs of V1 cells and CA1 place cells with overlapping firing fields.

We also performed the same analysis on two control groups of V1-CA1 cell pairs (see *Figure 6—figure supplement 2* for examples of V1-CA1 pairs). The first group consisted of 6681 pairs, each made of a CA1 place cell and a V1 location-responsive cell that both exhibited firing fields on a trajectory, but that their firing fields were non-overlapping. In this case, the Δrate and ΔCOM were computed within their most dominant firing fields on the trajectory. This control group, referred to as 'non-overlapping pairs', allowed us to test whether the CA1-V1 co-fluctuation was spatially confined within the overlapped firing fields. The second group contained 238 pairs, each composed of a CA1 cell that had place field on a trajectory and an active V1 cell that was not location-responsive on the trajectory. In this case, the Δrate and ΔCOM fluctuations of the V1 cell were computed within a spatial interval that was overlapped with the dominant CA1 place field (see 'Materials and methods'). We call this group 'non-responsive' pairs, which allowed us to test whether the co-fluctuation was specific to the location-responsive V1 and CA1 cells. We found that the overlapping pairs had

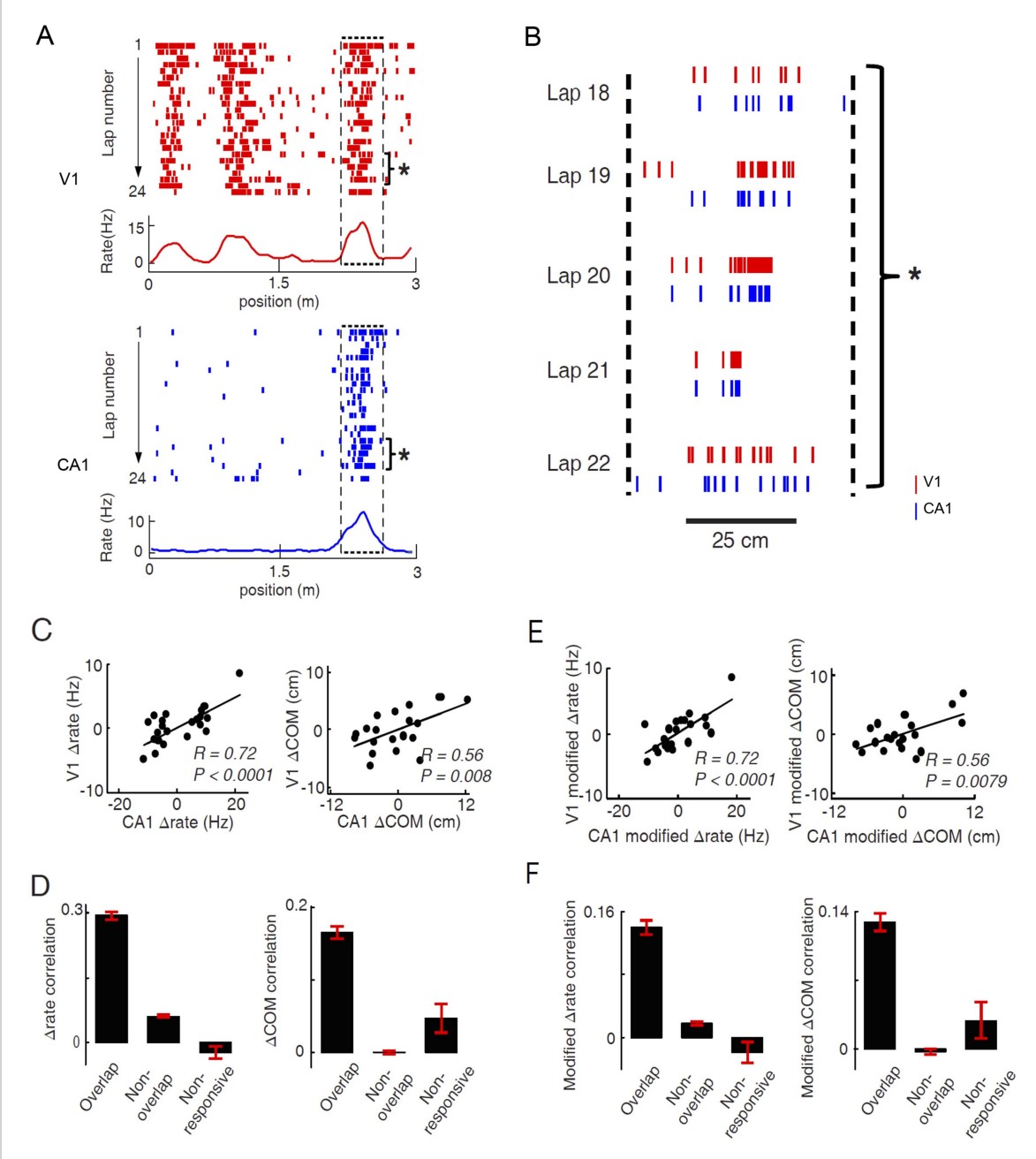

**Figure 6**. Pairs of V1 and CA1 cells with overlapping firing fields displayed correlated lap-by-lap fluctuations in firing rate and firing location. (**A**) Lap-by-lap spike raster and the average rate curves (see *Figure 2* legend for details) of a pair of V1 and CA1 cells on the same trajectory. *Boxes*: the overlapping firing fields of the two cells. (**B**) The spikes within the marked laps (*) of the two cells in A are expanded and plotted together. Note the correlated lap-by-lap shifting of the V1 and CA1 cells in their spikes. (**C**) The lap-by-lap fluctuations in firing rate (ΔRate) and COM (ΔCOM) within the firing fields of the two CA1 and V1 cells in A. Each dot is a lap. *Solid line*: linear regression. *R*, *P*: Pearson's correlation between the CA1 and V1 fluctuations and the associated p-value. (**D**) Average correlation in Δrate and ΔCOM for overlapping, non-overlapping, and non-responsive pairs of CA1 and V1 cells (see text for definitions). (**E**, **F**) Same as C and D, but for modified Δrate and modified ΔCOM after removing the modulations of firing rate and COM by speed and head direction.

The following figure supplements are available for figure 6:

**Figure supplement 1**. Two more examples of overlapping V1-CA1 cell pairs with correlated lap-by-lap fluctuations in, each from a different animal.

**Figure supplement 2**. Illustration of overlapping, non-overlapping, and non-responsive V1-CA1 cell pairs.

*Figure 6. continued on next page*

*Figure 6. Continued*

**Figure supplement 3**. Overlapping V1-V1 and CA1-CA1 cell pairs displayed correlated lap-by-lap fluctuation in firing rate and COM within their firing fields.

**Figure supplement 4**. Modulation of V1 and CA1 firing activities by speed and head direction.

**Figure supplement 5**. V1 location-responsive cells showed much less lap-by-lap backward shift in their firing locations than CA1 place cells.

significantly higher correlation in $\Delta$rate than both non-overlapping (correlation: $0.062 \pm 0.003$; $p < 0.0001$, $t$-test) and non-responsive pairs ($-0.023 \pm 0.015$; $p < 0.0001$; *Figure 6D*). Similarly, the overlapping pairs had significantly higher correlation in $\Delta$COM than non-overlapping (correlation: $0.0005 \pm 0.0028$; $p < 0.0001$) and non-responsive pairs (correlation: $0.047 \pm 0.020$; $p < 0.0001$; *Figure 6D*). We also computed the $\Delta$rate and $\Delta$COM correlations for CA1-CA1 cell pairs and for V1-V1 cell pairs. The results were similar: Overlapping pairs within each of the two brain areas were significantly correlated in both $\Delta$rate and $\Delta$COM and had significantly higher correlation than non-overlapping pairs and non-responsive pairs (*Figure 6—figure supplement 3*). Taken together, the results above demonstrate a specific, precise co-fluctuation in the firing rates and COMs of V1 location-responsive cells and CA1 place cells with overlapping firing fields, suggesting a functional interaction between these cells. Remarkably, this long–range interaction between cells in the distal V1 and CA1 was qualitatively similar to the local interaction within CA1 place cells.

Next, we examined whether the co-fluctuation of activity in overlapping V1-CA1 cell pairs could be explained by the lap-by-lap behavioral fluctuations. As expected (*Huxter et al., 2003*; *Saleem et al., 2013*), the firing rate and COM were modulated by speed and head direction in many V1 and CA1 cells (*Figure 6—figure supplement 4*). We quantified the modulation of both speed and head direction on $\Delta$rate/$\Delta$COM by a multi-variant linear regression and then removed the modulation to obtain the modified lap-by-lap fluctuations in firing rate/COM (modified $\Delta$rate/$\Delta$COM), which were no longer correlated with speed or head direction (see 'Materials and methods'). We computed the correlations in the modified $\Delta$rate/$\Delta$COM for overlapping V1-CA1 cell pairs. For the pair in *Figure 6A*, the correlations in their modified $\Delta$rate and $\Delta$COM remained unchanged (*Figure 6E*), although there was a modest reduction in other pairs (*Figure 6—figure supplement 1*). For the group of overlapping V1-CA1 pairs, the average correlation between the modified $\Delta$rate ($0.14 \pm 0.009$) remained significantly greater than 0 ($p < 0.0001$, $t$-test), and was significantly higher than those of non-overlapping ($0.018 \pm 0.002$, $p < 0.0001$) and non-responsive ($-0.019 \pm 0.014$, $p < 0.0001$) V1-CA1 pairs (*Figure 6F*). Similarly, the average correlation between the modified $\Delta$COM ($0.13 \pm 0.009$) was also significantly greater than 0 ($p < 0.0001$), and was significantly higher than those of non-overlapping ($-0.003 \pm 0.003$, $p < 0.0001$) and non-responsive ($0.029 \pm 0.018$, $p < 0.0001$) V1-CA1 pairs (*Figure 6F*). From these results, we conclude that, behavioral variations cannot fully account for the correlation in the firing rate and COM of overlapping V1-CA1 cell pairs.

The co-fluctuation of V1-CA1 cells suggests that those overlapping pairs of V1-CA1 cells with both displaying bidirectional firing should show the same type of directionality (prospective or retrospective firing), similarly as in previous reports on CA1 cells and entorhinal grid cells (*De Almeida et al., 2012*; *Bieri et al., 2014*). Our data yielded 4 such overlapping and 10 non-overlapping pairs of V1 and CA1 cells. We found that all the 4 overlapping pairs showed the same type of directionality, but only 5 of the 10 non-overlapping pairs did so whereas the other 5 showed the opposite directionality. Although the number of such bi-directional overlapping pairs is low (mainly due to the low percentage of bidirectional V1 and CA1 cells), the result is consistent with the correlated fluctuation in the firing rate and firing location between overlapping V1 and CA1 cells. In addition, CA1 cells are known to systematically shift their COMs backward lap by lap (*Mehta et al., 1997*, *2000*), presumably as a result of synaptic plasticity (*Ekstrom et al., 2001*). To understand whether the observed co-fluctuation in COM between V1 and CA1 cells were related to this lap-by-lap plastic change, we examined the lap-by-lap shift in the COMs of V1 cells. Our analysis shows that, although there was a sign of backward shifting in V1 cells during the first 5 laps or so, the shift was not as robust as and much smaller than that of CA1 cells (*Figure 6—figure supplement 5*). The result suggests that V1 firing activities are less plastic than those of CA1 cells at this

short time scale of laps (*McClelland et al., 1995*), and that the co-fluctuation between V1 and CA1 cells is not primarily driven by rapid plasticity in their firing activities.

## Experience dependence of V1 firing activities

In our experiments, V1 and CA1 cells were recorded as the animals ran the same track for multiple days. We next analyzed how the activities of V1 location-responsive cells and CA1 place cells changed over the many days' experience of track running. We grouped the recording days to 3 different time points (T1–T3): T1 included Day 1 and 2, a novel condition to the animals, in which most of the behavioral changes occurred (*Figure 1B,C*), T3 included Day 6 to Day 7+, which we consider to be a familiar condition, and T2 included Days 3–5, an intermediate condition between novel and familiar. We examined 279 V1 and 628 CA1 cells at T1, 173 V1 and 519 CA1 cells at T2, and 212 V1 and 557 CA1 cells at T3 that were active on at least one trajectory. First, we found that there was a significant increase from T1 to T3 in the median overall firing rate of V1 cells (50% increase between T1 and T3; $p < 0.0001$, *Kruskal–Wallis test* including all data at T1, T2 and T3), but not in the median firing rate of CA1 cells (3% increase between T1 and T3; $p = 0.079$; *Figure 7A*). Second, there was a significant increase in the median SMI for both V1 (55% increase between T1 and T3; $p < 0.0001$) and CA1 cells (39% increase between T1 and T3; $p < 0.0001$; *Figure 7B*), suggesting an experience-dependent increase in the location-specificity of V1 cells at the time scale of days. Third, we examined the correlation in Δrate and ΔCOM between overlapping V1-CA1 cell pairs at different time points ($N = 244$ pairs at T1, 108 pairs at T2, 174 pairs at T3). For the Δrate correlation, there was no significant change from T1 to T3 (2% decrease between T1 and T3; $p = 0.20$, *one-way ANOVA* comparing all data at T1, T2 and T3; *Figure 7C*). This is also true for the modified Δrate correlation after the speed and head direction modulation was removed (19% increase between T1 and T3; $p = 0.13$; *Figure 7D*). For the ΔCOM correlation, there was a small, but statistically non-significant decrease from T1 to T3 (17% decrease between T1 and T3; $p = 0.11$, *one-way ANOVA*; *Figure 7C*). For the modified ΔCOM, the decrease was more prominent, but remained non-significant (33% decrease between T1 and T3; $p = 0.088$; *Figure 7D*). These data show that, although there was an indication of slightly decreased co-fluctuation between V1 and CA1 cells with experience, their functional interaction persisted as the animals became familiarized with the track. Taken together, day-to-day track experience was accompanied by an increase in the firing rate and location-specificity of V1 cells, which remained correlated in their lap-by-lap fluctuations with overlapping CA1 cells.

## Layer differences in V1 firing activities

Finally, we asked whether the firing properties of V1 neurons differed across layers. We determined the laminar locations of the recorded neurons on the final recording day by performing small electrolytic lesions and subsequently examining the Nissl and acetylcholinesterase (AChE) staining (*Figure 1D*). We estimated the locations of those neurons recorded on previous days by relative distances tetrodes had traveled. A neuron's location was assigned to either the superficial layers 2 and 3 (L2/3), layer 4 (L4), or the deep layers 5 and 6 (L5/6). We obtained 478 cells in L2/3, 120 in L4, and 83 in L5/6 that were active on a trajectory. A small number of V1 cells (95 out of 776, 12%) were not assigned to a layer due to failed AChE staining. We first analyzed the overall firing rate and SMI among the cells recorded in different layers (*Figure 8A,B*) on all days. There was a significant difference in firing rate across the layers ($p < 0.0001$, *Kruskal–Wallis* test). The post-hoc pair-wise comparisons revealed that L4 cells had significantly higher firing rates (10.2 [6.5 14.9] Hz) than both L2/3 (3.2 [1.7 7.0] Hz, $p < 0.0001$, *ranksum* test) and L5/6 (2.9 [1.8 6.3] Hz, $p < 0.0001$) cells, whereas the latter two were similar ($p = 0.67$). The analysis of SMI shows that 86% of L2/3, 78% of L4, and 70% of L5/6 cells were location-responsive. The results indicate that L4 cells had the highest firing rate, whereas L2/3 cells had the highest percentage of location-responsive cells. Second, we computed the correlations in Δrate and ΔCOM for overlapping pairs of CA1 cells with the cells in each of the three cortical locations (CA1-L2/3, CA1-L4, and CA1-L5/6). We found that L2/3, L4 and L5/6 cells were similarly correlated in Δrate and ΔCOM with CA1 cells (data not shown). Third, we have shown that V1 cells increased their firing rates and SMIs with track running experience (T1 to T3, *Figure 7A,B*). Here we examined whether this was the case for cells in all layers of V1. We found a significant increase in firing rate only in L2/3 cells ($p = 0.00023$, *Kruskal–Wallis* test comparing all data at T1, T2 and T3; 39% increase between T1 and T3), but not in L4 ($p = 0.60$; 2.5% increase between T1 and T3) or L5/6 cells

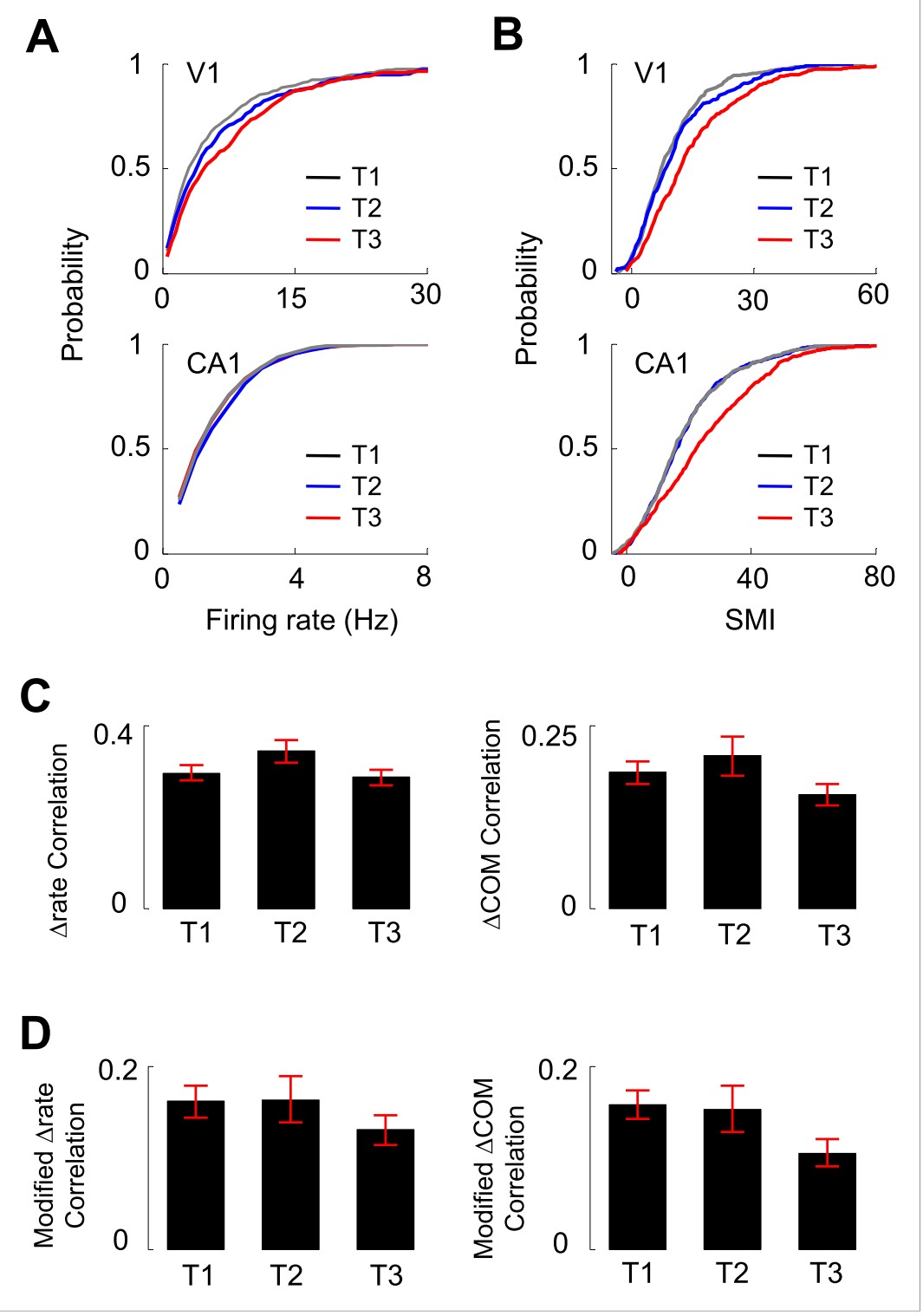

**Figure 7**. Experience dependence of V1 and CA1 firing activities on the C-shaped track. (**A**, **B**) Cumulative distributions of overall firing rate (**A**) and SMI (**B**) of active V1 and CA1 cells on different days (T1 – T3, see texts for definition). (**C**) The average (mean ± S.E.) correlation in the lap-by-lap Δrate (*left*) and ΔCOM (*right*) fluctuations for overlapping V1-CA1 cell pairs on different days. (**D**) Same as **C**, but for the correlation in modified Δrate and ΔCOM after removing the modulation by speed and head direction.

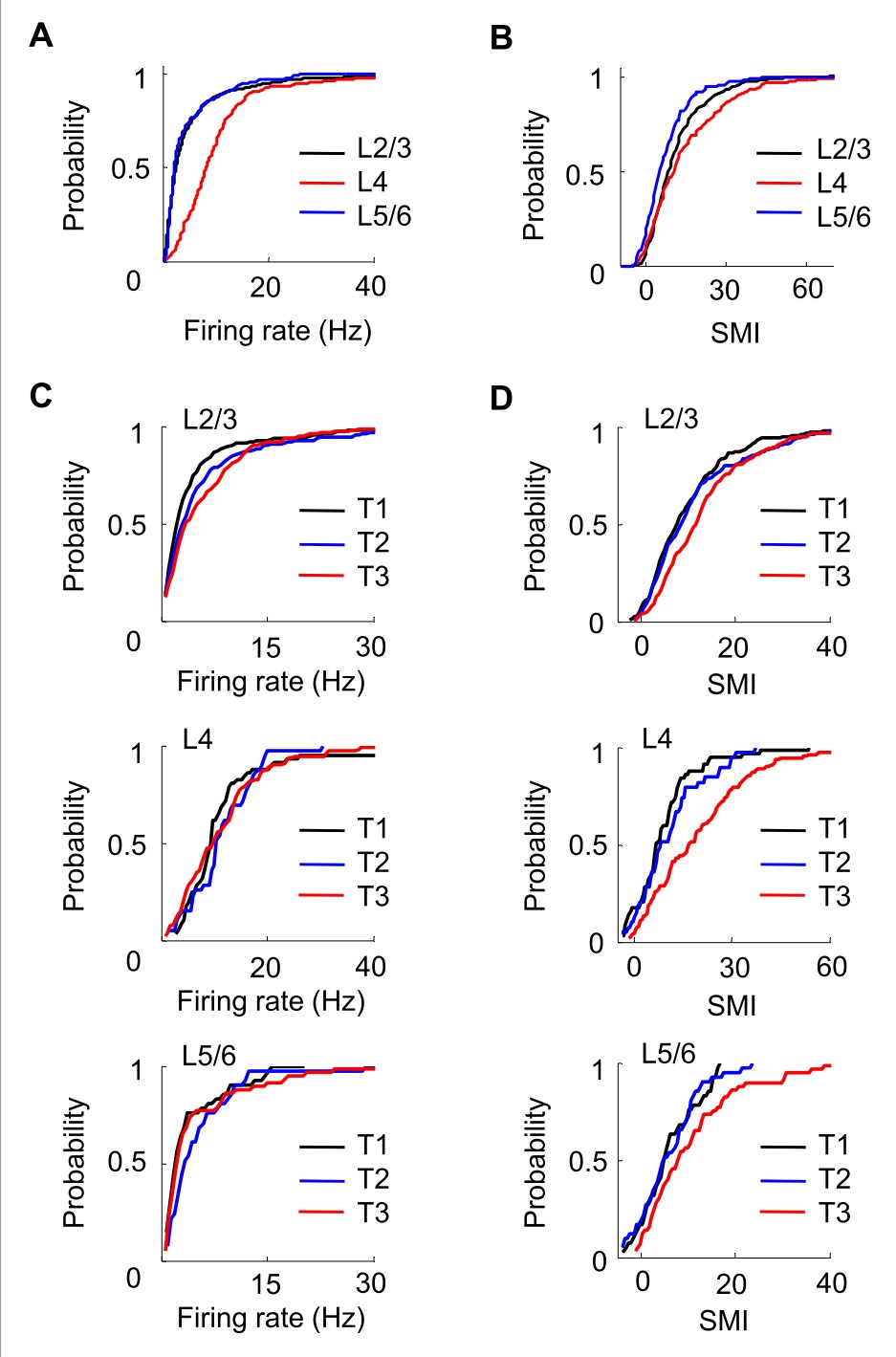

**Figure 8**. Layer differences in V1. (**A**, **B**) Cumulative distributions of overall firing rate (**A**) and SMI (**B**) for active cells in L2/3, L4 and L5/6. (**C**, **D**) Cumulative distributions of firing rate and SMI for L2/3, L4, and L5/6 cells on different days (T1–T3).

(p = 0.10; 2.3% increase between T1 and T3; *Figure 8C*). However, there was a significant increase in SMI from T1 to T3 for cells in all layers (L2/3: p < 0.0001, 52% increase between T1 and T3; L4: p < 0.0001, 130% increase between T1 and T3; L5/6: p = 0.05, 54% increase between T1 and T3; *Figure 8D*).

# Discussion

We have analyzed the firing activities of V1 and CA1 cells in freely moving rats as they ran back and forth on a C-shaped track for food reward. We found that a large number of cells in all layers of V1, like CA1 place cells, fire selectively and reliably at specific locations of the track. These location-responsive V1 cells have multiple, narrow firing fields on a given trajectory of the track. The number of V1 firing fields rises and falls as animals approach and move away from visual landmarks, leading the rise and fall in the number of CA1 place fields. The precise lap-by-lap fluctuations in the firing rate and firing location of V1 cells are correlated with those of CA1 cells with overlapping firing fields. Finally, V1 cells display an experience-dependent change in their location-specificity at the time scale of days. These results reveal that V1 cells have location-specific responses to a particular environment and are functionally coupled with the hippocampal place cells encoding the same environment.

Our data supports a functional interaction between V1 location-responsive cells and hippocampal place cells. Here we provide three pieces of evidence. First, the number of V1 firing fields fluctuates as the animal travels through a trajectory and this fluctuation is highly correlated with the change in the number of CA1 firing fields. More interestingly, we found that the V1 fields rise and fall earlier than the CA1 fields and this leading effect is consistent on both running directions. Second, the pair-wise cross-correlation analysis reveals a bias toward positive time lags, suggesting that, again, V1 location-specific activity appears earlier than the CA1 place field activity at the population level. Third, the lap-to-lap fluctuations in the firing rates and firing locations of individual V1 cells within their firing fields are significantly correlated with those of CA1 place cells with overlapping place fields. This is true even after removing the modulation of speed and head direction on firing rate and firing location. The correlated variation is specific to CA1-V1 cell pairs with the overlapping firing fields, suggesting a remarkably precise interaction at a fine temporal and spatial scale. These observed correlations are unlikely driven by non-specific common factors because of the following reasons. First, both the field distribution and pair-wise cross-correlation analyses show that V1 activities lead CA1 activities during trajectory running. Although the lead could be caused by CA1 and V1 cells independently responding to some unknown common factors with a delay, which we cannot rule out, the lead time of 60–120 ms, is consistent with our previous studies on the interaction between V1 and CA1 during slow-wave sleep (*Ji and Wilson, 2007*; *Haggerty and Ji, 2014*), when behavioral factors do not vary and the brain state is relatively uniform. Therefore, the delay likely reflects the propagation of activity from V1 to CA1. Second, the lap-by-lap co-fluctuation of overlapping pairs is much higher than that of non-responsive pairs. Since both types of pairs share overlapping locations, if common factors were mainly responsible, the non-responsive pairs would show correlated fluctuations comparable to those of overlapping pairs.

The leading of V1 activity over CA1 activity suggests a forward interaction, through which V1 location-responsive cells send visual information to CA1 place cells. The location-specific activity in V1 cells does not necessarily mean that V1 cells directly respond to places per se as hippocampal place cells do. Given what we know about the primary function of V1 neurons in behaving rodents (*Yao et al., 2007*; *Niell and Stryker, 2010*; *Xu et al., 2012*), they are likely driven by the visual cues associated with specific locations on the track, supported by our observation that V1 firing fields are more distributed around the landmarks. In particular, prospective firing may occur when V1 neurons 'see' landmarks in front of the animal. Retrospective firing may occur as V1 neurons 'see' landmarks behind the animal's head, which is possible because the side-facing eyes of rats have a wide visual field and can see visual stimuli behind or above the head (*Wallace et al., 2013*), although other mechanisms such as feedback from the higher cortical areas or the hippocampus may also be responsible. The landmark information represented by the V1 cells can reach the hippocampus via a multi-synaptic pathway, which includes the secondary visual cortex, temporal cortex, perirhinal/postrhinal cortex, and entorhinal cortex (*Miller and Vogt, 1984*; *Vaudano et al., 1991*; *Lavenex and Amaral, 2000*; *Furtak et al., 2007*). In this way, the V1 location-responsive cells may allow local and distal visual cues to influence CA1 place cell activities (*Muller and Kubie, 1987*; *Lee et al., 2004a*, *2004b*; *Leutgeb et al., 2005*).

Our finding of location-responsive cells in the sensory area V1 raises the possibility that the distinction between what we call either a spatial or sensory response is not as clear as we thought. On one hand, given the apparent modulation of hippocampal place cells by visual cues (*Muller and Kubie, 1987*; *Lee et al., 2004a*, *2004b*; *Leutgeb et al., 2005*), the response of a hippocampal cell to

a particular place also contains a 'sensory' response to the visual cues associated with that place. In our data, the CA1 place fields are preferentially distributed around the landmarks, just as the V1 firing fields are, suggesting a bias of CA1 place cell activity by sensory input. On the other hand, the response of a V1 cell to visual clues at a particular place effectively establishes a 'spatial' response to the place, if the visual cues are stable in space. In this manner, the firing fields of V1 cells may appear to be similar to the CA1 place fields, with the important differences in that V1 cells exhibited less specificity by nature of non-zero baseline firing and multiple firing fields. The difference in V1 and CA1 location-specific activity is just that, because hippocampal place cell activities result from an integration of not just visual information, but also information from other sensory modalities and self-motion, hippocampal place fields are more specific and predictive of the animal's spatial location than the predominantly sensory-driven V1 firing fields. Therefore, it is possible that spatial representation in the brain may involve a gradual transformation along the anatomical pathway connecting the sensory cortex to the hippocampus, which converts a primarily sensory-driven, less reliable spatial response in the sensory cortex to a more precise and robust spatial response in the hippocampus.

Hippocampal place cells are believed not just to form a neural representation of space, but encode spatial memory and other type of memories (*Eichenbaum et al., 1999*; *Burgess and O'Keefe, 2003*; *Ferbinteanu et al., 2006*; *Knierim at al., 2006*; *Moser et al., 2008*). If so, then the gradual transformation of spatial representation also suggests that a spatial memory trace is not only encoded by hippocampal place cells, but also by the cortical cells involved in the transformation, which are presumably distributed in many cortical areas. This idea is an essential component in many modern theories of memory, including index theory (*Teyler and Rudy, 2007*), two-stage memory processing theory (*Buzsaki, 1996*), and the theory of two complementary memory systems (*McClelland et al., 1995*). All these theories propose the involvement of relevant cortical cells in long-term memory storage. Our previous study shows that activity patterns of V1 and CA1 cells during track running are coordinately replayed later during slow-wave sleep (*Ji and Wilson, 2007*), suggesting the involvement of V1 cells in memory consolidation. The current study provides another piece of key evidence that V1 cells and CA1 place cells functionally interact during the stage of memory formation as animals run a novel track for a first week or so. In our data, the lap-by-lap co-fluctuation between V1-CA1 cell pairs with overlapping firing fields is qualitatively similar to that between pairs of CA1-CA1 place cells with overlapping place fields. Since CA1 place cells with overlapping place fields are believed to constitute functional 'cell assemblies' that encode the spatial memory of the animal's environment (*Harris et al., 2003*; *Dragoi and Buzsaki, 2006*), it is possible that the overlapping V1 location-responsive cells and hippocampal place cells are part of the larger 'cortical-hippocampal assemblies' that encode and store the same spatial memory trace. Therefore, we propose that the V1 location-specific activities are a substrate for encoding and storing the visual component of long-term spatial memories, and possibly episodic memories.

## Materials and methods

### Animals and behavioral procedure

Fifteen Long-Evans rats, all male, 3–6 months old, were used in the electrophysiological recording experiments. The rats were first food deprived to ≥85% of their *ad libitum* weight and pre-trained for 5–10 days to run back and forth on a ~2 m long linear track for milk reward delivered at both ends of the track. The rats were then placed back onto an *ad libitum* diet and went through a surgery for implanting a tetrode recording device (see below). About 4 weeks after the surgery, animals were once again food deprived and recording began as the animals were placed on a novel C-shaped track (*Figure 1A*), in a different room from the pre-training. The track, made of metal, was ~3 m long and ~6 cm wide with 3 cm high walls. The track was surrounded by a black curtain. No additional visual cues were added to the experimental set up. The only local cues were those inherent to the C-shaped metal track with 4 corners and low walls (*Figure 1A*). During the recording, rats ran back and forth along 2 trajectories for milk remotely delivered at both ends of the track by syringe and tubing from outside the curtain. The animals were free to move along the track, but rewarded only after they reached one end from the other. The recording started on the very first day the animals experienced the track (Day 1) and continued for 2–14 days, with 20–60 min each day. The experimental protocol was approved by the Institutional Committee on Animal Care at Baylor College of Medicine and followed National Institutes of Health guidelines.

## Surgery

For each rat, we implanted a tetrode recording device (tetrode drive), containing 24 independently movable tetrodes made of 4 fine nichrome wires (diameter 13 μm), under isoflurane anesthesia. Twelve tetrodes were implanted into an exposure at coordinates anteroposterior (AP) −3.8 mm, mediolateral (ML) 2.5 mm from Bregma and the other 12 at AP -6.5 mm and ML 4.8 mm, for targeting CA1 and V1, respectively. The tetrode drive was fixed to the skull with stainless anchoring screws and dental cement. The analgesic ketoprofen (5 mg/kg) was administrated by subcutaneous injection before the animal was allowed to recover from the anesthesia.

## Tetrode recording

Tetrode recordings, by using a Digital Lynx system (Neuralynx, Bozeman MT), were performed as previously described (*Ji and Wilson, 2007*, *2008*). Briefly, during the ~4 weeks post surgery, tetrodes were slowly moved to the CA1 and V1. Once stable single units (spikes presumably from single neurons) in CA1 were obtained, recordings started on Day 1 when the animals were introduced to the track and performed the track running task as described above. Once the recordings started on Day 1, CA1 tetrodes were never moved again on later recording days, but V1 tetrodes were moved no more than 125 μm a day to sample cells in different layers of V1. Some V1 tetrodes were already moved to deep layers before the recording started, in order to sample deep-layer cells on early days of track running. Signals recorded by each tetrode were filtered with a pass band of 600 Hz–9 kHz. Spikes were identified by a pre-set threshold of 50–70 μV and digitized at a sampling rate of 32 kHz. Two color diodes (red, green) were mounted over the animal's head to track the animals' position and head direction. Positions were sampled at 33 Hz with a resolution approximately 0.2 cm. Rat position was analyzed offline, *x* and *y* position was linearized, and animal speed was calculated as the linearized spatial distance between each 2 time points.

## Histology

Animals were euthanized using pentobarbital overdose (≥200 mg/kg) after the tetrode recording. For each tetrode, an electric current (30 μA) was passed for ~15 s to create a small lesion at the tip of the tetrode. Brains were fixed in 10% formalin for at least 24 hr and then sectioned at 50 μm thickness. The sections were alternatively stained with either 0.2% Cresyl violet or 1% sodium sulfide nonahydrate ($Na_2S_9H_2O$) to detect acetylcholineesterase (AChE) activity following the established protocol (see *Paxinos and Watson, 2007* for details in procedure and the agents used). The lesion sites were verified from the lesion marks in the stained sections (*Figure 1D*), according to the standard rat brain atlas (*Paxinos and Watson, 2007*). At the implantation coordinates used here, area V1 is flanked laterally and medially by the secondary visual cortex V2. V1 was delineated from surrounding cortical areas by a granulized layer 4 that exhibited denser staining in the Cresyl violet and the presence of a double band of darker AChE stain restricted to V1 in layers 4 and 5. The presence of two dark AChE bands and the darker layer 4 Cresyl staining were also used to locate the laminar location of the lesion sites within V1 (*Zilles et al., 1984*). Because CA1 tetrodes were never moved during the recording days, the locations of all CA1 cells were identified by the lesion sites. The locations of V1 cells recorded on the final recording day were also the same as the lesion sites in V1. The locations of V1 cells recorded on previous days were estimated, based on the distances tetrodes traveled to the final locations. In two rats, the AChE staining procedure did not work. V1 neurons in these animals were not assigned to a layer.

## Data analysis

Single units were sorted using custom software (xclust, M. Wilson at MIT, available at GitHub repository: https://github.com/wilsonlab/mwsoft64/tree/master/src/xclust) on all data recorded across all recoding days. Since some tetrodes ware not moved from one day to another (but could have drifted slightly overnight), it is difficult to know whether different or same cells were recorded across days from these tetrodes. Therefore, certain cells might be repeatedly sampled across days, but were included in the analysis because we would like to examine how V1 and CA1 activities changed across days (*Figures 7* and *8*). A total of 852 V1 and 3627 CA1 cells were recorded. For CA1 cells, only putative pyramidal cells (firing rate <5 Hz) that were active on at least one trajectory (firing rate ≥0.5 Hz) were included in the analysis (*Ji and Wilson, 2007*, *2008*). For V1 cells, all active cells

(firing rate >0.5 Hz) were included. We did not exclude high-rate V1 cells because we did not find reliable indicators that high-rate V1 cells were a distinct cluster from low-rate V1 cells in our data. This criterion yielded a total of 776 V1 cells and 2033 CA1 cells across all the recording days, which were the basis for all of the analyses. Only the activities of these cells during active track running were analyzed. The activities at the food wells (the final ~10 cm at each end of the track) or during the stopping periods on the track (with running speed <6 cm/s for at least 0.5 s) were excluded from the analysis. This exclusion is due to the fact that our purpose was to study the location responses of CA1 and V1 cells, but it is known that during stopping behavior CA1 cells fire in a different mode from active running and their activities become non-location-specific (*Foster and Wilson, 2006*; *Diba and Buzsaki, 2007*; *Cheng and Frank, 2008*; *Karlsson and Frank, 2009*). Results were expressed as median [25–75%] range values, mean ± S.E. values, or as otherwise specified. Accordingly, *ranksum* and *Kuskal-Wallis* tests were used for statistical comparisons with median values, *t-test* and *ANOVA* for comparisons with mean values, or as specified otherwise. p-values of the statistical tests were reported as exact values unless <0.0001.

## Firing rate curve and spatial information measures

We defined two running trajectories on the C-shaped track as the two paths from one end to the other in opposite running directions (*Figure 1A*). Each time an animal traveled along a trajectory, it was defined as a lap on the trajectory. Each trajectory was linearized and divided into 2 cm spatial bins. If a cell was active on both trajectories, the cell was analyzed separately on each trajectory. The firing rate curve of a cell was the cell's firing rate at each bin of a trajectory: $[x_1, x_2, ..., x_N]$. The rate $x_i$ at the $i$-th bin was the total number of spikes across all laps divided by the total time the animal spent at the bin (occupancy time, $t_i$). The firing rate curve was smoothed by a Gaussian kernel with a sigma of 2 bins. Spatial information content ($SI_c$) was defined as (*Skaggs et al., 1993*),

$$SI_c = \sum_{i=1}^{N} p_i \frac{x_i}{r} \log_2 \frac{x_i}{r}, \tag{1}$$

where $p_i = \frac{t_i}{\sum_{i=1}^{N} t_i}$ was the occupancy probability, $r = \sum_{i=1}^{N} p_i x_i$ was the mean firing rate, and $N$ was the number of bins. Spatial information rate ($SI_r$) was (*Skaggs et al., 1993*),

$$SI_r = SI_c \times r. \tag{2}$$

## Spatial stability

Given any two laps of a trajectory, suppose the cell's firing rates at the $i$-th bin were $x_i$ and $y_i$. Then, the correlation coefficient between these two laps' spatial rate curves was:

$$C = \frac{1}{N} \sum_{i=1}^{N} \left( \frac{x_i - \overline{x}}{\sigma_x} \right) \left( \frac{y_i - \overline{y}}{\sigma_y} \right), \tag{3}$$

where $\overline{x}$ and $\sigma_x$ are the mean and standard deviation of $[x_1, x_2, ..., x_N]$ and $\overline{y}$ and $\sigma_y$ are the mean and standard deviation of $[y_1, y_2, ..., y_N]$, respectively. We computed $C$ values for all combinations of laps and took their mean as the spatial stability for the cell on the trajectory (*Cheng and Ji, 2013*).

## Random shuffling and spatial modulation index (SMI)

To compare the spatial response in a cell's spiking activity with the chance level, we shuffled the spiking activity of the cell during a lap on a trajectory by random, circular shifting (*Henriksen et al., 2010*; *Igarashi et al., 2014*). Suppose the [start end] times of a lap were $[T_1\ T_2]$ and a cell had a spike train of M spikes within the lap at times $\boldsymbol{P} = [P_1, P_2, ..., P_M]$. We generated a random number $R$ in $[0, T_2-T_1]$. A random spike train for this lap was $T_1 + \{[(\boldsymbol{P} + R)-T_1] \bmod (T_2-T_1)\}$. The random number was independently generated for each lap. A complete shuffled version of the cell's spiking activity was thus composed of all the shuffled laps' random spike trains on a trajectory. This shuffling procedure preserved the firing rate and even the temporal spiking pattern, but disrupted the spatial correlates of the spikes. For computing the SMI of a cell, we shuffled its spiking activity 100 times and then computed the $SI_c$ (or equivalently $SI_r$) value for each shuffle (*Figure 2—figure supplement 1*).

Let $m$, $s$ denote the mean and standard deviation of all the 100 randomly generated $SIc$ values and $a$ the actual $SIc$ value of the cell. Then, $SMI$ was the z-score of $a$: $SMI = (a−m)/s$. A cell was considered as location-responsive if $SMI > 2.325$, which corresponded to the 99th percentile of the chance level.

## Identification of firing fields

For a cell's firing rate curve on a trajectory $[x_1, x_2, …, x_N]$, we defined a baseline firing rate as the 30$^{th}$ percentile of $[x_1, x_2, …, x_N]$. For cells with relatively low firing rate such as CA1 cells and most V1 cells in L2/3 and L5/6, this baseline rate was close to 0. This definition of baseline was empirical. All the results in later analyses were qualitatively similar and only quantitatively affected if the baseline was defined as between 15$^{th}$ and 50$^{th}$ percentile. We subtracted the baseline from the firing rate curve and identified the peaks of the remaining curve. If a peak of the remaining curve was greater than 1 Hz and greater than 20% of the baseline rate, we counted it as a firing field and identified its boundaries as 10% of the peak rate (*Mehta et al., 2000*). If the gap between two neighboring fields was smaller than 4 cm, the two fields were merged. This analysis was performed only on trajectory-active CA1 cells and on trajectory-active V1 cells identified as location-responsive.

## Spatial and temporal cross-correlations, field distribution and bi-directionality

For computing firing field distribution, we counted the number of V1 or CA1 field peaks within each bin of 2 cm along each of the two trajectories on the track to produce an histogram (*Figure 3A,B*). The counts were then smoothed by a Gaussian window with a sigma of 2 bins. Cross-correlograms between firing field distribution curves (*Figure 3*) or between firing rate curves (*Figure 4*) were computed similarly as in *Equation 3* at each position lag of 2 cm. A cell was considered bi-directional if it was location-responsive on the two opposite trajectories and its firing rate curves on the two trajectories had a cross-correlogram with a significant peak location within [−50, 50] cm (Pearson's $p < 0.00098$, adjusted down from 0.05 to account for multiple tests at all 51 position lags). Cross-correlation between two cells' spiking activities (*Figure 5*) was computed as a normalized spike count correlation. Given two spike trains made of $M$, $N$ spikes, $P = [P_1, P_2, …, P_M]$ and $Q = [Q_1, Q_2, …, Q_N]$, and a time bin size $B$, for each time lag $\Delta T$ and for each spike $P_i$ ($i = 1, 2, ..., M$) in $P$, we counted the number of spikes in $Q$ that were within $[P_i+\Delta T-B/2$ $P_i+\Delta T + B/2]$. The cross-correlation $C(\Delta T)$ between the cells at time lag $\Delta T$ was the sum of the spike counts for all the $i = 1, 2, ..., M$. The cross-correlation $C(\Delta T)$ was normalized by the spike count that would be expected if the two spike trains were random Poisson events, which was $m = M*N/B$. That is $C(\Delta T) = (C(\Delta T)–m)/\sqrt{m}$. Similar to previous studies (*Sirota et al., 2003*; *Siapas et al., 2005*), this normalized spike count correlation is not sensitive to firing rate. The cross-correlogram was the cross-correlation vs the time lag. $B$ was set as 10 ms. We defined a highly significantly correlated pair of cells as the one with a peak normalized cross-correlation greater than a threshold of 8 ($p < 0.0001$, z-test). The result in *Figure 5D* was similar for a range of threshold values between 3 and 10.

## Correlation in lap-by-lap firing rate and COM fluctuations

We computed the correlation in the lap-by-lap firing rate and COM fluctuations between pairs of cells with overlapping firing fields. An overlapping pair was defined as a pair of cells that (i) at least one firing field of a cell overlapped spatially with a field of another cell by at least 50%, and (ii) their cross-correlogram had a dominant peak with peak correlation ≥8 and peak time within [−200, 200] ms. The criterion in the cross-correlogram was to exclude those pairs with only a minor portion of spikes fired within their overlapping fields. For a given cell pair with overlapping firing fields on a trajectory and for each lap, we identified the spikes of each cell within each cell's corresponding overlapped field. If a pair had multiple overlapped firing fields, we only analyzed the spikes within the most dominant pair of overlapping fields (the one with the maximum sum of peak rates). We then computed the firing rate and the mean firing location (center of mass—COM) of the spikes for each cell and for each lap within the overlapped firing fields. For each cell, the firing rate/COM fluctuation (Δrate/ΔCOM) for a lap was the difference between the lap's firing rate/COM and the average firing rate/COM across all the laps. Suppose the lap by lap Δrate/ΔCOM of one cell was $x = [x_1, x_2, …, x_N]$ and those of another were $y = [y_1, y_2, …, y_N]$, where $N$ was the number of laps, their correlation in Δrate/ΔCOM was given by *Equation 3*.

We also applied the analysis to non-overlapping cell pairs and non-responsive cell pairs (*Figure 6—figure supplement 2*). A pair of cells was included in the non-overlapping pairs if they

both had firing fields on a trajectory, but if none of their firing fields overlapped by at least 50%. In this case, the lap-by-lap firing rate and COM were computed within their dominant firing fields (the one with the highest peak rate). The non-overlapping pairs served as a control group to test whether the correlated fluctuation was spatially confined within overlapping firing fields. A pair of cells was considered a non-responsive pair if one cell was location-responsive and had place fields on a trajectory while the other was not location-responsive. The non-responsive pairs served as a control group to test whether the correlated fluctuation was limited to location-responsive cells. For non-responsive pairs, the lap-by-lap fluctuations were computed within the dominant firing field for the location-responsive cell, and for the non-responsive cell within a randomly generated spatial interval that was overlapped with the field by a random percentage between 50%and 100%.

## Correlation between Δrate/ΔCOM with behavioral parameters

For each cell on a trajectory, we computed running speed and head direction for each lap the animal traveled through its dominant firing field. The head direction at each lap was subtracted from the circular mean direction of all laps (Δhdir). For each cell, we then computed the correlation between the lap-by-lap speed (or Δhdir) and the lap-by-lap Δrate (or ΔCOM). Although head direction was a circular variable, we still used Pearson' correlation (*Equation 3*) to compute the correlation between Δhdir and Δrate (or ΔCOM), because the Δhdir were mostly within $[-30°, 30°]$ (*Figure 6—figure supplement 4*).

## Correlation in modified Δrate/ΔCOM

Suppose the lap-by-lap Δrates (or ΔCOM), speeds, and Δhdirs of a cell were $x = [x_1, x_2, …, x_N]$, $V = [V_1, V_2, …, V_N]$, and $D = [D_1, D_2, …, D_N]$, respectively, where $N$ was the number of laps. We first performed a multi-variant linear regression: $x = \alpha V + \beta D + x'$. The regression residual $x'$ was the lap-by-lap modified Δrate (or ΔCOM). By definition, $x'$ had 0 correlation with $V$ or $D$. Therefore, this procedure removed any modulation of speed and head direction on firing rate and COM. We then repeated the correlation in fluctuation analysis for overlapping, non-overlapping and non-responsive pairs by replacing $x$ with $x'$. Removing the modulation of speed and head direction this way did not always reduce correlation. Occasionally, it increased the correlation if the two cells in a pair were oppositely modulated by speed or head direction (e.g., if one increased, but the other decreased firing rate as head direction turned from left to right).

# Acknowledgements

This work is supported by a grant (EY020676) from the National Eye Institute to DJ, and supported in part by the Baylor College of Medicine IDDRC cores (grant number 1U54 HD083092 from the Eunice Kennedy Shriver National Institute of Child Health & Human Development).We thank A Tolias and Ji lab members for discussions.

# Additional information

## Funding

| Funder | Grant reference | Author |
| --- | --- | --- |
| National Eye Institute (NEI) | R01 EY020676 | Daoyun Ji |

The funder had no role in study design, data collection and interpretation, or the decision to submit the work for publication.

## Author contributions

DCH, Acquisition of data, Analysis and interpretation of data, Drafting or revising the article; DJ, Conception and design, Analysis and interpretation of data, Drafting or revising the article

## Ethics

Animal experimentation: The experimental procedures in this study were approved by the Institutional Committee on Animal Care at Baylor College of Medicine (Protocol #5134) and followed National Institutes of Health guidelines.

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
