## [Decision Letter]

Thank you for submitting your work entitled “Activities of visual cortical and hippocampal neurons co-fluctuate in freely moving rats during spatial navigation” for peer review at *eLife*. Your submission has been favorably evaluated by Timothy Behrens (Senior Editor), a Reviewing editor, and two reviewers.

The following individuals responsible for the peer review of your submission have agreed to reveal their identity: Howard Eichenbaum (Reviewing Editor) and Francesco Battaglia (peer reviewer).

The reviewers have discussed the reviews with one another and the Reviewing editor has drafted this decision to help you prepare a revised submission

This paper reports on data from joint electrophysiological recordings of hippocampal CA1 and V1 ensembles, during shuttling on a C-shaped track. The authors report that V1 cells have spatially related responses, and that their activity seems to be related to the spatially-selective activity of CA1 neurons. The main arguments supporting the link between visual cortex and hippocampal activity are a similar distribution of “place field” peaks, the existence of fine scale temporal correlations captured by the cross-correlogram, and the co-variability of firing rate and field center of mass across runs. This is a very interesting study that builds upon an earlier study by [27] that showed place-specific firing in V1 that was coordinated with place cell firing in CA1 during waking and sleep. Although the present findings are not as novel as the authors imply, due to some overlap with the previous Ji and Wilson study, there is still much new and interesting information provided in this paper. Nonetheless, there are some issues that should be resolved prior to publication.

Reviewer #1:

1) In the first paragraph of the subsection “Location-specific firing activities of V1 cells”, the authors write: “Interestingly, many V1 cells also dramatically increased their firing rates at specific locations of a trajectory”. This wording makes it sound as though the authors were surprised by this finding. However, this should have been expected, considering that stable, location-specific firing was reported previously by [27]. The authors need to present their current results in the context of these earlier results.

2) The authors' hypothesis regarding the recordings from the different layers of V1 was not entirely clear, and the results were also described in a way that seemed a bit strange to me. In the second paragraph of the subsection “Location-specific firing activities of V1 cells”, the authors report that “cells in layer 4 had the highest firing rates and cells in deep layers had least SMI”. However, the authors do not mention that cells in layer 4 had the highest SMI values, and this result is not explained/discussed. The deep layers result is as I would have expected, but I would have expected layers 2 and 3, not layer 4, to exhibit the highest SMI values.

3) The authors state that the non-responsive pairs served as a control group to test whether the correlated fluctuation was limited to location-responsive cells. But, the authors compared a randomly selected interval in V1 with the place field interval in CA1 to assess fluctuations for the non-responsive pairs. I did not understand why the authors did this because one would not expect to see correlated fluctuations when comparing different time intervals. Wouldn't it be better to use the same interval in the non-responsive V1 cell as was used in CA1?

4) As seen in Figure 3, it appears as though the highest number of firing fields occurred near reward sites, yet this is not mentioned.

5) The authors report prospective and retrospective firing in V1 cells and CA1 cells. Did V1 cells and CA1 cells tend to exhibit prospective firing and retrospective firing at the same time, as has been reported by [8] for ensembles of grid cells, and by [2] for ensembles of place cells? This is implied by the findings in Figure 6, but perhaps it could be tested directly.

6) In the subsection “Temporal correlation between V1 and CA1 firing activities”, the authors write: “…when a V1 cell and a CA1 cell both had well-defined firing fields on the same trajectory, they often had a normalized cross-correlogram with a prominent peak”. This language is a bit vague. What exactly is meant by “often”? How many V1 and CA1 cell pairs had well-defined firing fields on the same trajectories and what percentage of cell pairs had a prominent peak in the cross-correlogram?

7) I did not understand the following passage: “…prospective firing may occur when landmarks fall into the receptive fields of V1 cells in front of the animal's head direction on both running directions, and retrospective firing occurs when landmarks fall into the receptive fields from behind the animal's head position.” How can something be in front of a direction? And, how can landmarks fall into receptive fields from behind the animal's head position?

8) In the subsection “Data analysis”, the authors write: “Therefore, certain cells might be repeatedly sampled across days, but were included in the analysis because we would like to examine how V1 and CA1 activities changed during the learning of the task across days”. This sounds fine, but was a repeated measures design then used to analyze the data (i.e., using an identifier to indicate which tetrode locations or cells were repeated across days)?

9) In the subsection “Random shuffling and spatial modulation index (SMI)”, I did not understand what the authors meant by “a copy” in the following passage: “A copy of random spiking activity for the cell was the random spike trains for all the laps on a trajectory”.

10) In the Methods, the authors do not describe how running speed was calculated.

Reviewer #2:

In general, what is missing from the paper is some element of dissociation between V1 spatial and visual responses. The only salient visual landmarks (standing to the author's interpretation, and we would agree on that) are the two 90 degrees bends in the track. Some moveable landmarks would have helped a great deal in helping to understand the data.

I understand that that cannot be fixed, but I still see many interesting things in the paper, especially about cortico-hippocampal exchange, and those may still be enhanced in a proper revision.

I have an issue with the use of the SMI measure for comparing different populations, as done in the subsection “Location-specific firing activities of V1 cells”. This is essentially a z-scored spatial information value, compared with a circularly shifted surrogate dataset. The measure is perfectly fine in my opinion to determine if an information score is significant or not, but when comparing different populations, the denominator (the variance in the surrogate data) may be different, just due to the different structure of the firing rate map. In any event, the result is quite inconclusive, and I suggest removing it.

---

## [Author Response]

Reviewer #1:

*1) In the first paragraph of the subsection “Location-specific firing activities of V1 cells”, the authors write: “Interestingly, many V1 cells also dramatically increased their firing rates at specific locations of a trajectory”. This wording makes it sound as though the authors were surprised by this finding. However, this should have been expected, considering that stable, location-specific firing was reported previously by*
[27]*. The authors need to present their current results in the context of these earlier results*.

We agree and have amended the text.

*2) The authors' hypothesis regarding the recordings from the different layers of V1 was not entirely clear, and the results were also described in a way that seemed a bit strange to me. In the second paragraph of the subsection “Location-specific firing activities of V1 cells”, the authors report that “cells in layer 4 had the highest firing rates and cells in deep layers had least SMI”. However, the authors do not mention that cells in layer 4 had the highest SMI values, and this result is not explained/discussed. The deep layers result is as I would have expected, but I would have expected layers 2 and 3, not layer 4, to exhibit the highest SMI values*.

We agree with the reviewer that the sentence did not summarize the layer difference accurately. In light of the reviewer #2’s comment (see below), comparing SMIs across different cell populations is problematic, but using SMI to detect the percentage of location-specific cells in a population is valid. Therefore, following reviewer #2’s suggestion, we have dropped the SMI comparison across the layers, but kept the percentage of location-specific cells in each layer. We found 86% of layer 2/3 cells, 78% of layer 4 cells, and 70% of layer 5/6 cells contained significant spatial information, which makes more sense in a way that more cells in the output layer (layer 2/3) are location-specific than the input layer (layer 4) and the feedback layer (layer 5/6). However, we feel like we really do not have a clear idea why these cells are distributed this way. Although this is not a key result and does not affect our conclusion, it is a useful piece of information for readers and future studies (we have also expanded this analysis to address the reviewer #2’s comment below). Therefore, we report the result as a matter of fact (in the subsection “Layer differences in V1 firing activities” and in a new Figure 8) without further elaboration.

3) The authors state that the non-responsive pairs served as a control group to test whether the correlated fluctuation was limited to location-responsive cells. But, the authors compared a randomly selected interval in V1 with the place field interval in CA1 to assess fluctuations for the non-responsive pairs. I did not understand why the authors did this because one would not expect to see correlated fluctuations when comparing different time intervals. Wouldn't it be better to use the same interval in the non-responsive V1 cell as was used in CA1?

We think that the phrase “randomly chosen field” may have contributed to this confusion. For a pair of V1 and CA1 cells in this control group, we did use a spatial interval of the V1 cell that overlapped with the place field of the CA1 cell. However, to make the degree of the overlap similar to that of overlapping pairs, this spatial interval was not always exactly the same as the CA1 place fields, but shifted randomly to keep the overlap within 50% – 100%, which was the criterion for the overlapping pairs. We have modified the wording in the Results (subsection “Lap-by-lap co-fluctuation of V1 and CA1 firing activities “) to remove the confusion. A more detailed description is provided in the Materials and methods section (second paragraph of subsection “Correlation in lap-by-lap firing rate and COM fluctuations”) and Figure 6—figure supplement 2.

*4) As seen in*
Figure 3*, it appears as though the highest number of firing fields occurred near reward sites, yet this is not mentioned*.

We agree that this may appear to be the case from Figure 3. We have, therefore, added a sentence (subsection “Spatial correlation between V1 and CA1 firing activities”) suggesting a high place field density close to the reward sites. We did not probe the place fields exactly at the reward sites, because we excluded the activities at the reward sites, namely the final ∼10 cm at each end of the track, from the analysis of firing fields. This is due to the fact that the animals were mostly stationary (eating and resting), which is a different behavioral state from running and triggers non-place firing of CA1 cells (as in ripple oscillations). We have also revised the related sentences in the Materials and methods section to clarify this (subsection “Data analysis”).

*5) The authors report prospective and retrospective firing in V1 cells and CA1 cells. Did V1 cells and CA1 cells tend to exhibit prospective firing and retrospective firing at the same time, as has been reported by*
[8]
*for ensembles of grid cells, and by*
[2]
*for ensembles of place cells? This is implied by the findings in*
Figure 6*, but perhaps it could be tested directly*.

We agree with the reviewer that this is an important analysis, which we have performed. We obtained pairs of V1 and CA1 cells with overlapping firing fields (overlapping pairs) and those with non-overlapping firing fields (non-overlapping pairs) and then asked whether the two cells in any given pair showed the same or different type of directionality (prospective or retrospective firing). However, we only obtained 4 overlapping pairs and 10 non-overlapping pairs with both cells showing bi-directionality. The low numbers of pairs are due to the low percentage of bidirectional cells (∼10% of V1 and CA1 cells) in our data. Nevertheless, all the 4 overlapping pairs showed same type of bi-directionality, whereas half of the non-overlapping pairs showed the same type and the other half showed the opposite. We also tried to relax our definition of overlapping pairs (by lessening the required percentage of overlap), but did not obtain substantially more pairs. Although the analysis is not conclusive due to low numbers of pairs, it is consistent with the results of our co-fluctuation analysis. We have added this new result to the text in the Results section with a cautious tone (subsection “Lap-by-lap co-fluctuation of V1 and CA1 firing activities”).

*6) In the subsection “Temporal correlation between V1 and CA1 firing activities”, the authors write*: *“…when a V1 cell and a CA1 cell both had well-defined firing fields on the same trajectory, they often had a normalized cross-correlogram with a prominent peak”. This language is a bit vague. What exactly is meant by “often”? How many V1 and CA1 cell pairs had well-defined firing fields on the same trajectories and what percentage of cell pairs had a prominent peak in the cross-correlogram?*

We have adjusted the text to remove the ambiguous word “often” and provided the number and percentage of pairs with a significant peak within the time interval [-0.2 0.2] s.

7) I did not understand the following passage: “…prospective firing may occur when landmarks fall into the receptive fields of V1 cells in front of the animal's head direction on both running directions, and retrospective firing occurs when landmarks fall into the receptive fields from behind the animal's head position.” How can something be in front of a direction? And, how can landmarks fall into receptive fields from behind the animal's head position?

We regret a typo (“the animal’s head direction” should be “the animal’s head position”) in this sentence. What we assumed in the sentence is that rats have side-facing eyes with a wide view field (>270⁰). Therefore, a V1 cell can “see” a landmark before the animal reaches it, which could lead to prospective firing, and even after the animal’s head passes it, which could lead to retrospective firing. However, there may be other possibilities like feedbacks from higher order areas including the hippocampus. Therefore, we have re-worked the sentences here (Discussion, third paragraph) to make these points clearer.

8) In the subsection “Data analysis”, the authors write: “Therefore, certain cells might be repeatedly sampled across days, but were included in the analysis because we would like to examine how V1 and CA1 activities changed during the learning of the task across days”. This sounds fine, but was a repeated measures design then used to analyze the data (i.e., using an identifier to indicate which tetrode locations or cells were repeated across days)?

We agree with the reviewer that a repeated measure design would be better. However, an inherent problem with the electrophysiological recording is that it is very difficult to determine whether the same cells or different cells were recorded from the same electrode/tetrode across days. If a tetrode slightly drifts overnight, as it often does, without 24-hour continuous tracking, it is hard to know whether the cells recorded on two consecutive days are the same or different. Therefore, we feel uncomfortable to use an identifier to indicate which cells were repeatedly measured across days. But rather, we treated cells recorded on different days in a given area/layer as longitudinal samples from the same population with replacements. This treatment is a more conservative way of analysis, which does not take advantage of the repeated measure design in, e.g., ANOVA, and tends to yield a less significant p value. The analysis mainly applies to the experience-dependent changes of V1 and CA1 cells (Figure 7). If anything, it only underestimates the significance of the differences across days. Since we also report the effect size (percentage of change), this underestimate does not affect our results in any substantial way.

*9) In the subsection “Random shuffling and spatial modulation index (SMI)”, I did not understand what the authors meant by “a copy” in the following passage: “A copy of random spiking activity for the cell was the random spike trains for all the laps on a trajectory”*.

We have rearranged the text here to clarify the procedure.

10) In the Methods, the authors do not describe how running speed was calculated.

We have added a sentence indicating the method by which running speed was calculated (“Rat position was analyzed offline, x and y position was linearized, and animal speed was calculated as the linearized spatial distance between each 2 time points”).

Reviewer #2:

*In general, what is missing from the paper is some element of dissociation between V1 spatial and visual responses. The only salient visual landmarks (standing to the author's interpretation, and we would agree on that) are the two 90 degrees bends in the track. Some moveable landmarks would have helped a great deal in helping to understand the data*.

*I understand that that cannot be fixed, but I still see many interesting things in the paper, especially about cortico-hippocampal exchange, and those may still be enhanced in a proper revision*.

We agree on the point that an experiment which can dissociate V1 spatial and visual responses would significantly enhance the study. However, our focus here is to systematically characterize the V1 responses in freely moving animals with a maze-running task and determine the possible V1-hippocampal interactions and their dependence on experience, which have not been examined in previous studies. We believe the results provide valuable insights into how V1 cells respond during natural behavior of freely moving animals and how they interact with hippocampal place cells.

*I have an issue with the use of the SMI measure for comparing different populations, as done in the subsection “Location-specific firing activities of V1 cells”. This is essentially a z-scored spatial information value, compared with a circularly shifted surrogate dataset. The measure is perfectly fine in my opinion to determine if an information score is significant or not, but when comparing different populations, the denominator (the variance in the surrogate data) may be different, just due to the different structure of the firing rate map. In any event, the result is quite inconclusive, and I suggest removing it*.

Thinking over the reviewer’s argument, we realize that comparing SMIs across different cell populations with very different firing characteristics needs caution. Following the reviewer’s suggestion, we have removed the direct comparisons in SMI between V1 and CA1 cells and between different layers of V1 cells. However, we think it is still legitimate to compare the SMIs of same group of cells across different days (Figures 7 and 8), given that the basic nature of firing rate maps should not change dramatically in same cells across days.
